# The role of energy storage in deep decarbonization of electricity production

Maryam Arbabzadeh[1], Ramteen Sioshansi[2], Jeremiah X. Johnson[3] & Gregory A. Keoleian [1]

Deep decarbonization of electricity production is a societal challenge that can be achieved with high penetrations of variable renewable energy. We investigate the potential of energy storage technologies to reduce renewable curtailment and $CO_2$ emissions in California and Texas under varying emissions taxes. We show that without energy storage, adding 60 GW of renewables to California achieves 72% $CO_2$ reductions (relative to a zero-renewables case) with close to one third of renewables being curtailed. Some energy storage technologies, on the other hand, allow 90% $CO_2$ reductions from the same renewable penetrations with as little as 9% renewable curtailment. In Texas, the same renewable-deployment level leads to 54% emissions reductions with close to 3% renewable curtailment. Energy storage can allow 57% emissions reductions with as little as 0.3% renewable curtailment. We also find that generator flexibility can reduce curtailment and the amount of energy storage that is needed for renewable integration.

[1] Center for Sustainable Systems, School for Environment and Sustainability, University of Michigan, Ann Arbor, 48109 MI, USA. [2] Department of Integrated Systems Engineering, The Ohio State University, Columbus, 43210 OH, USA. [3] Department of Civil, Construction, and Environmental Engineering, North Carolina State University, Raleigh, 27607 NC, USA. Correspondence and requests for materials should be addressed to M.A. (email: marbab@umich.edu)

D ue to cost decreases[1,2], renewable energy is experiencing greater use (https://www.eia.gov/outlooks/steo/pdf/steo_full.pdf). Many jurisdictions have policies in place to incentivize renewable use (http://www.dsireusa.org/). These policies are often intended to decrease the carbon-intensity of electricity production.

The role of energy storage in aiding the integration of renewable energy into electricity systems is highly sensitive to the renewable-penetration level[3]. California, for instance, is experiencing days during which demand is too low to accommodate all of the solar energy that is available midday[4]. This overgeneration-related renewable curtailment can be exacerbated by thermal generators having limited flexibility in how quickly they can adjust their production or how low their production levels can go[5].

The development and deployment of grid-scale energy storage is advancing due to technology development and policy actions, such as California's energy storage mandate[6,7]. Energy storage can provide a variety of services and its economic rationale is highly application-dependent[8]. Numerous studies optimize the size and operation of energy storage within a specific power system to achieve the best economic or environmental outcome. However, there are no studies in the extant literature that investigate systematically the economic viability of using energy storage to alleviate renewable curtailment for the purposes of decarbonizing electricity production. Moreover, the existing literature does not examine the impacts of emissions policy, such as a carbon tax, on the economics of energy storage for mitigating renewable curtailment. Detailed analysis is required to estimate the value of energy storage that is used for different applications, including renewable integration[9]. This study addresses this gap by optimizing the investment in and operation of nine currently available energy storage technologies to minimize cost of the California and Texas power systems. We assume varying renewable penetrations and different $CO_2$-tax policies.

Energy storage technologies have different characteristics and potential applications[10–13]. As such, no single technology excels on all characteristics. Integrating energy storage into the grid can have different environmental and economic impacts, which depend on performance requirements, location, and characteristics of the energy storage system[14–16]. The cost of energy storage systems and regulatory challenges are major obstacles to their adoption[13,17–19]. Braff et al.[20] examine the value of using energy storage to increase the price at which wind and solar energy can be sold in wholesale markets. They find that many energy storage technologies are currently too costly for this application and determine the cost reductions that are needed to make this application economically viable. Other works[21–25] examine the environmental impacts of energy storage, showing that it depends upon how it is operated and the technical characteristics of the power system into which it is integrated.

Thus, there is a need to optimize the operation of energy storage to achieve desired economic and environmental outcomes. Many studies optimize the operation and size of an energy storage system for a given grid application based on economic criteria[26,27]. Others propose optimization models for sizing and operating energy storage to minimize total electricity cost or to maximize investor profits[28–30]. Another set of studies model emissions and economic considerations in optimizing energy storage use[31–33].

Our study extends the existing literature by evaluating the role of energy storage in allowing for deep decarbonization of electricity production through the use of weather-dependent renewable resources (i.e., wind and solar). The model optimizes the power and energy capacities of the energy storage technology in question and power system operations, including renewable curtailment and the operation of generators and energy storage. This is done to minimize total system costs, which consist of the capital cost of energy storage, generator-operations costs, and $CO_2$-emissions costs. Technical constraints in the model include operating limits of generators and energy storage and load-balance requirements. We examine nine currently available energy storage technologies: pumped-hydroelectric storage (PHS), adiabatic (ACAES), and diabatic (DCAES) compressed air energy storage (CAES), and lead-acid (PbA), vanadium-redox (VRB), lithium-ion (Li-ion), sodium-sulfur (NaS), polysulfide bromide (PSB), and zinc-bromine (ZNBR) batteries. Our model allows us to determine which energy storage technologies are most cost-effective in aiding renewable integration and the extent to which the cost of a currently uneconomic technology must come down to make it cost-effective. We use two case studies, which are based on the California and Texas power systems in 2010–2012, and consider up to 20 GW of wind and 40 GW of solar capacity added to the system. We also consider the impact of a $CO_2$ tax of up to $200 per ton.

Our analysis of the cost reductions that are necessary to make energy storage economically viable expands upon the work of Braff et al.[20], who examine the combined use of energy storage with wind and solar generation assuming small marginal penetrations of these technologies. Conversely, we examine their economics at significant renewable penetrations that are necessary for deep decarbonization of electricity production.

Our findings show that renewable curtailment and $CO_2$ reductions depend greatly on the capital cost of energy storage. Moreover, increasing the renewable penetration or $CO_2$ tax makes energy storage more cost-effective. This is because higher renewable penetrations increase the opportunities to use stored renewable energy to displace costly generation from non-renewable resources. Among the energy storage technologies that we consider, PHS and DCAES are deployed in more of the scenarios that we examine. This is due to the lower capital costs of these technologies. Other technologies see deployment under some scenarios. We also find that relatively modest reductions in the capital costs of other energy storage technologies can make them cost-effective for this proposed application.

## Results

**Energy storage deployment.** Supplementary Table 1 summarizes the energy capacity of the energy storage technologies that are installed with different wind- and solar-penetration levels and $CO_2$ emissions-tax regimes in 2012 in the base case with a 7.0-GW minimum-dispatchability requirement in the California Independent System Operator (CAISO) system. Supplementary Table 2 summarizes the same for the Electric Reliability Council of Texas (ERCOT) system under the base-case 8.2-GW minimum-dispatchability requirement. The tables show that higher renewable penetrations or emissions taxes tend to improve the economics of energy storage deployment. Due to their relatively low capital costs, PHS and DCAES are deployed in more scenarios and with greater capacity than most of the other technologies. Conversely, a technology that is currently more-expensive but has a higher round-trip efficiency, such as Li-ion batteries, is not deployed in any of the scenarios that are summarized in these two tables. Table 1 shows the results of a sensitivity analysis, in which lower cost assumptions for Li-ion batteries lead to significant Li-ion deployment and resultant curtailment and emissions reductions. Supplementary Data 1 summarizes the amounts of energy storage that are installed in the other years and with the other minimum-dispatchability requirements that we analyze.

**Table 1 Changes in renewable curtailment and $CO_2$ emissions resulting from lower Li-ion and PSB costs and higher NaS costs as a percentage of renewable curtailment and $CO_2$-emissions levels with the baseline costs**

| Technology | Renewable Curtailment (%) | | $CO_2$ Emissions (%) | |
|---|---|---|---|---|
| | CAISO | ERCOT | CAISO | ERCOT |
| Li-ion | −27.1 | −56.9 | −22.3 | −1.6 |
| NaS | 60.2 | 89.6 | 43.9 | 1.4 |
| PSB | −15.1 | −34.4 | −8.9 | −0.5 |

Results shown are for 2012 assuming 20-GW wind- and 40-GW solar-penetration level, a $200 per ton $CO_2$-emissions tax, and base-case minimum-dispatchability requirements of the CAISO and ERCOT systems

Supplementary Tables 1 and 2 show that irrespective of the carbon-tax level, energy storage is not cost-effective in California for the application that we model without added renewables. This is because California's fossil-fueled generators are all natural gas-fired. Thus, there is limited value in using energy storage for energy arbitrage, because of small differences between on- and off-peak marginal generation costs. In California, the value of energy storage stems primarily from its ability to reduce renewable curtailment, thereby displacing fossil-fueled generation. This benefit is greater with a higher carbon tax, because fossil-fueled generation is more costly in the presence of a tax. Recent estimates from the California Energy Commission show that as of October 2017, California has 5.6 GW of wind and 16.2 GW of solar installed (https://www.energy.ca.gov/almanac/renewables_data/wind/). Thus, California is approaching renewable-penetration levels at which a number of energy storage technologies are cost-effective for mitigating renewable curtailment.

Even in the absence of renewables, deploying some energy storage technologies in Texas is cost-effective under higher emissions-tax rates. This is because the ERCOT system has a more mixed generation fleet, with both coal- and natural gas-fired units that have very different generation costs. Moreover, the differences in the carbon contents of coal and natural gas gives larger differences in marginal generation costs between coal- and natural gas-fired units with higher $CO_2$-tax rates.

**Renewable curtailment**. Figure 1 shows total annual renewable curtailment with and without access to energy storage with different amounts of renewable capacity and $CO_2$-emissions taxes in 2012 in California under the base case 7.0-GW minimum-dispatchability requirement. Figure 2 shows the same for Texas under its 8.2-GW base case minimum-dispatchability requirement. The curtailment results for other minimum-dispatchability requirements and years are provided in Supplementary Data 1. The figures show that California has much higher renewable-curtailment rates relative to Texas. This is because California has much higher outputs from inflexible resources (e.g., nuclear, geothermal, biomass, and hydroelectric units) and energy imports compared to Texas. This greater inflexibility makes it more challenging for the CAISO system to absorb wind and solar generation. The figures show that with relatively low emissions taxes (i.e., $50 per ton or less), PHS and CAES are the only economically viable technologies for averting renewable curtailment. However, with higher emissions taxes, all of the energy storage technologies (except for Li-ion batteries) become cost-effective for this application. This is consistent with Supplementary Tables 1 and 2, which show that most of the energy storage

technologies are deployed in some of the renewable-penetration scenarios if the $CO_2$-emissions tax is sufficiently high.

Consistent with real-world experience[4], renewable curtailment is greatest in the spring. This is due to the spring having relatively low electricity demand and many days with good midday solar availability. California has experienced recently an increasing number of spring days on which these factors require solar curtailment.

**$CO_2$ emissions**. Figure 3 summarizes the benefits of energy storage in decarbonizing in-state electricity production in California in 2012, under the base case 7.0-GW minimum-dispatchability requirement. Figure 4 shows the same in Texas under the base case 8.2-GW minimum-dispatchability requirement. Results for other minimum-dispatchability requirements and years are provided in Supplementary Data 1. Without any added renewables or energy storage, California can achieve negligible 0.2% $CO_2$-emissions reductions with a sufficiently high carbon tax through dispatch switching. In Texas, dispatch switching can decrease emissions by 24% without added renewables. California's fossil-fueled generators have negligible emissions-rate differences. With a carbon tax, generating loads can be switched to units that have higher operating costs and lower emissions rates. Texas, conversely, has a mix of coal- and natural gas-fired generating units. A sufficiently high carbon tax switches the merit order between these units.

Without any access to energy storage, California's 2012 $CO_2$ emissions could have been reduced by 72%, through deployment of renewables with a 7.0-GW minimum-dispatchability requirement and a $200 per ton $CO_2$ tax. However, energy storage decarbonizes electricity production to a greater extent by reducing renewable curtailment. Li-ion batteries would have provided essentially no emissions improvements in 2012, due to their high capital costs. Conversely, DCAES yields the greatest emissions reductions in California in 2012. Texas shows similar trends. Without energy storage, renewable deployment, in conjunction with a $200 per ton $CO_2$-emissions tax, can reduce $CO_2$ emissions by 54% in 2012 with the base case 8.2-GW minimum-dispatchability level. As in California, DCAES yields the greatest emissions reductions in Texas.

Figure 5 summarizes energy storage's impacts on renewable curtailment and $CO_2$ emissions in California in the 3 years that we analyze. The results that are shown in the figure assume 20-GW and 40-GW wind- and solar-penetration levels, respectively, a $200 per ton $CO_2$-emissions tax, and the base-case 7.0-GW minimum-dispatchability requirement. The results are similar for other minimum-dispatchability requirements. Renewable curtailments are shown as percentages of potential renewable production while emissions reductions are reported as percentages relative to a no-renewables case. The figure shows significant interannual variability in renewable-curtailment rates, which stem from differences in electric loads. 2012 has significantly higher loads compared to 2010, meaning that California can accept more renewable generation in 2012. Each of the energy storage cases that is shown in the figure corresponds to the technology that achieves the greatest curtailment or emissions reduction. PHS achieves the greatest curtailment reductions in all of the years that are analyzed and the greatest emissions reductions in 2010. However, DCAES achieves greater emissions reductions in the other 2 years. These results suggest that if curtailment reduction is the goal of deploying energy storage, PHS is a relatively stable technology choice in California. Conversely, if emissions reduction is the policy priority, there is less technology robustness.

DCAES is, conversely, a more robust technology in Texas, achieving the greatest curtailment and emissions reductions in all

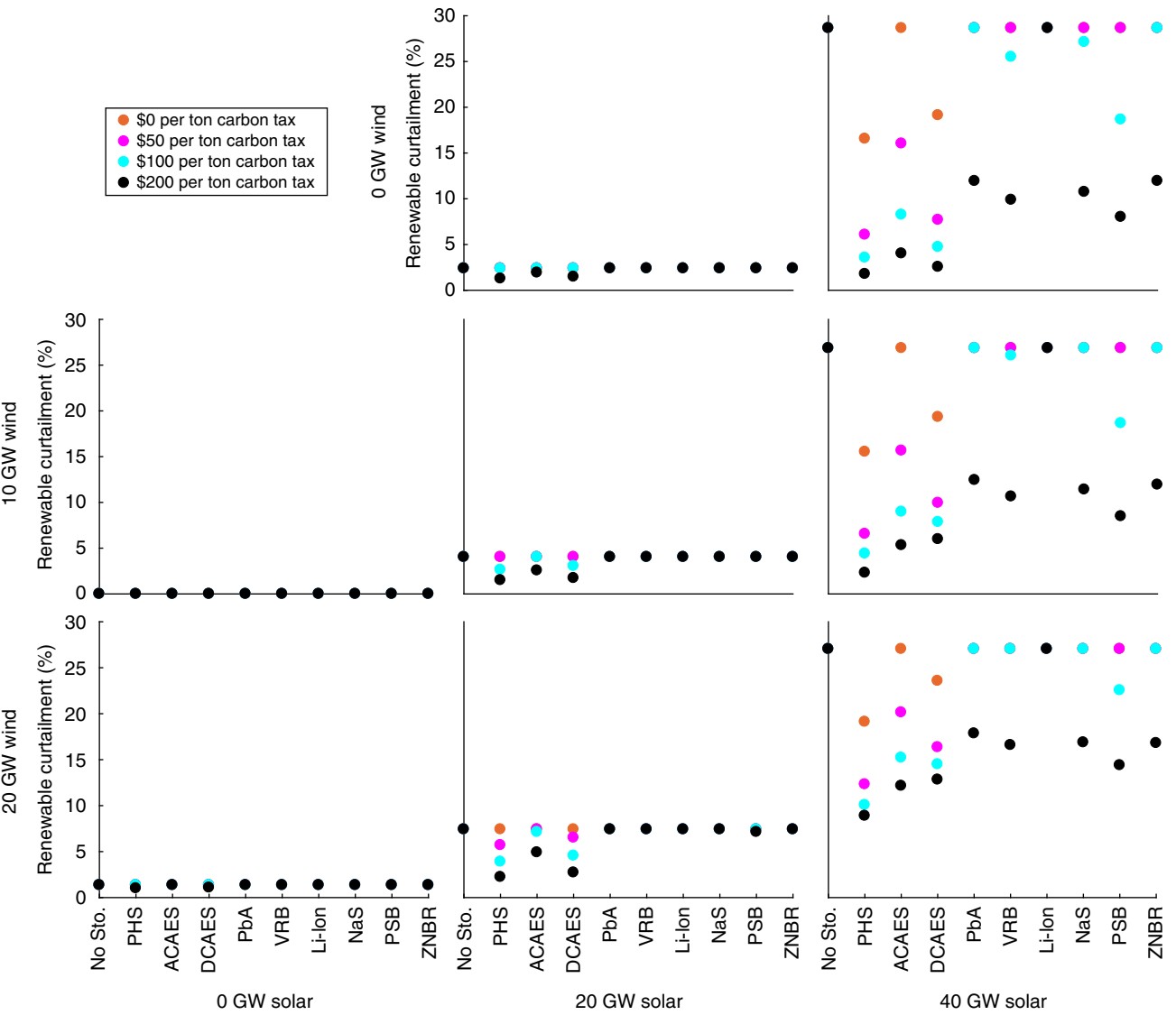

**Fig. 1** Annual renewable curtailment as a percentage of potential renewable production in California for the year 2012 in the base case with a 7.0-GW minimum-dispatchability requirement. Figure panels correspond to different wind- and solar-penetration levels, which are indicated at the left-hand side and bottom, respectively, of the figure. Source data are provided as a Source Data file

of the years and with all of the minimum-dispatchability requirements that we examine. However, energy storage delivers smaller incremental benefits in reducing Texas's $CO_2$ emissions. Figure 5 shows that without energy storage, adding 60 GW of renewables yields emissions reductions that range between 71 and 92% across the years that are analyzed. Energy storage increases these emissions reductions to between 90 and 97%. ERCOT achieves 52–56% emissions reductions from adding 60 GW of renewables without energy storage. DCAES increases these emissions reductions to 56–59%. This relatively small impact of energy storage in Texas is because there is relatively little renewable curtailment compared to California. As such, energy storage has a more limited role in increasing the use of renewable energy in Texas relative to California. Instead, the emissions-reduction benefits of DCAES in Texas largely stem from helping to shift some generating loads from coal- to natural gas-fired generators.

**Discussion**

Our case study shows that energy storage can play a non-trivial role in decarbonizing California's electricity production through greater use of renewables. Some technologies (e.g., PHS, CAES, and VRB and PSB batteries) can eliminate cost-effectively over 90% of $CO_2$ emissions relative to a no-renewables case. Without energy storage, massive renewable deployment can only achieve about 72% $CO_2$-emissions reductions (with the base-case 7.0-GW minimum-dispatchability requirement and a $200 per ton $CO_2$-emissions tax). In Texas, energy storage deployment yields 57% $CO_2$-emissions reductions compared to a no-renewables case (assuming an 8.2-GW minimum-dispatchability requirement and a $200 per ton emissions tax). Without energy storage, 60 GW of renewables reduce emissions by 54% relative to a no-renewables case. Recent analyses[1,34] show that Texas had over 22 GW of wind installed as of 2017. Thus, the case with 60 GW of renewables represents a significant increase in solar capacity and an already-achieved wind-penetration level.

California has less supply-side flexibility (i.e., more output from nuclear, geothermal, biomass, and hydroelectric units and energy transactions) compared to Texas, resulting in relatively high renewable curtailment in California. Thus, energy storage is valuable in reducing renewable curtailment and displacing fossil-fueled generation. Conversely, even without added renewables,

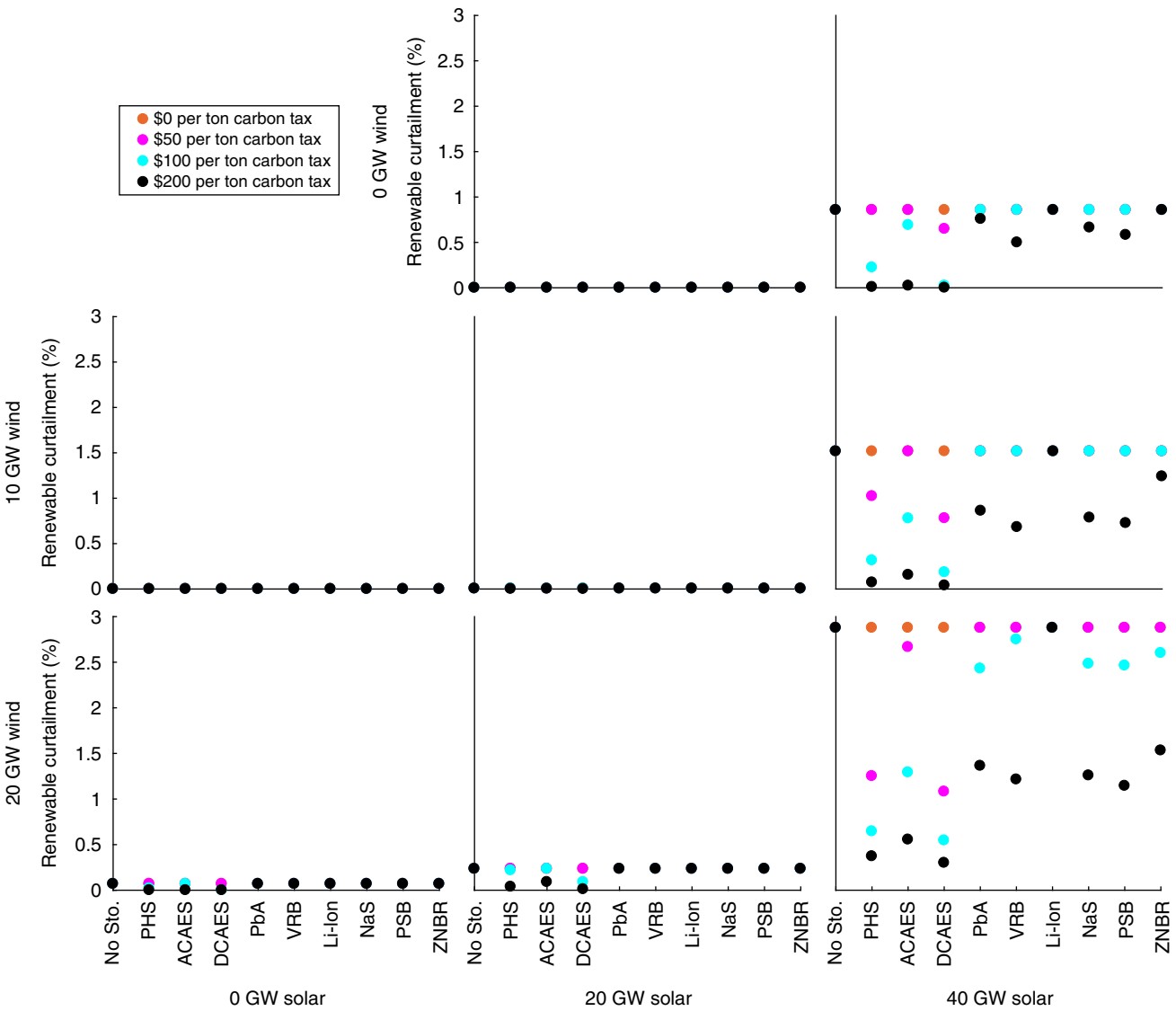

**Fig. 2** Annual renewable curtailment as a percentage of potential renewable production in Texas for the year 2012 in the base case with an 8.2-GW minimum-dispatchability requirement. Figure panels correspond to different wind- and solar-penetration levels, which are indicated at the left-hand side and bottom, respectively, of the figure. Source data are provided as a Source Data file

energy storage is cost-effective in Texas with a carbon tax, as it can be used to shift generating loads away from coal-fired units toward natural gas-fired generation.

Our results represent a lower bound on energy storage's role in renewable integration and electricity decarbonization. This is because at high renewable penetrations, energy storage may play other roles that are not captured in our model[3]. For instance, energy storage can be a low-cost source of flexibility to accommodate subhourly or minute-to-minute variability in wind and solar availabilities. Because our model assumes an hourly temporal resolution, such a benefit of energy storage is not captured.

Our results show that its capital cost is the primary factor in determining the scale at which an energy storage technology is deployed. Even with ambitious renewable penetrations and a high emissions tax, a relatively expensive (but high-efficiency) technology, such as Li-ion batteries, has a limited role to play. Our results suggest, however, that modest reductions in Li-ion-battery costs may increase their deployment. We determine this by examining the reduced cost of energy storage capacity, which is obtained from solving our optimization model. In the context of our model, the reduced cost can be interpreted as indicating how

much the capital cost of an uneconomic energy storage technology must change before it is cost-effective to build[35]. Our results show that in scenarios in which Li-ion batteries are not built, capital cost reductions of between $1 per kWh and $40 per kWh are sufficient to make the technology economically viable. Given the major reductions in battery-manufacturing costs over the past decade, such cost reductions may be possible. This would mean that energy storage technologies that appear uneconomic in our case study may well be viable in the near future. The reduced costs results for other storage technologies are provided in Supplementary Data 1.

Given the wide range of costs for Li-ion, NaS, and PSB batteries that are reported in the literature (https://www.lazard.com/media/450774/lazards-levelized-cost-of-storage-version-40-vfinal.pdf), we conduct a sensitivity analysis, in which the capital costs of Li-ion batteries are reduced to $259 per kWh and $59 per kW, the costs of NaS batteries are increased to $350 per kW and $350 per kWh, and the costs of PSB batteries are reduced to $200 per kW and $90 per kWh. Table 1 summarizes the impacts of these changed capital costs. Specifically, the table reports changes in renewable curtailments and $CO_2$ emissions relative to the levels

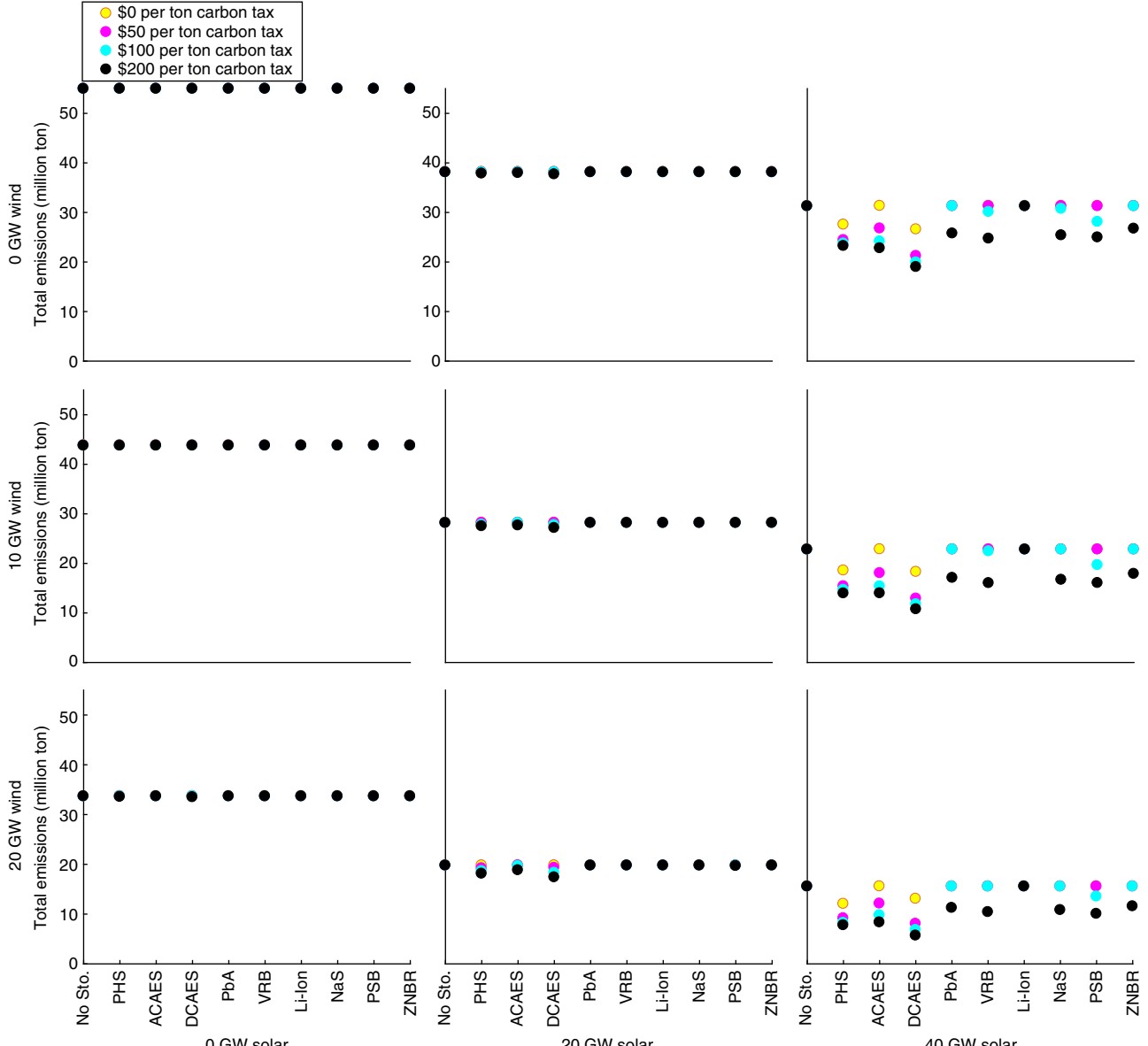

**Fig. 3** Annual $CO_2$ emissions (million ton) in California for the year 2012 in the base case with a 7.0-GW minimum-dispatchability requirement. Figure panels correspond to different wind- and solar-penetration levels, which are indicated at the left-hand side and bottom, respectively, of the figure. Source data are provided as a Source Data file

that are achieved with the baseline costs, as a percentage of the baseline curtailment and emissions impacts of Li-ion, NaS, and PSB. The results that are in the table are for 2012, assuming 20 GW of wind and 40 GW of solar are added to each system, a $200 per ton $CO_2$-emissions tax, and the base-case minimum-dispatchability requirement for each system. The amounts of energy storage added, renewable curtailments, and $CO_2$ emissions that are achieved in other scenarios are provided in Supplementary Data 1.

Our results demonstrate that increasing the $CO_2$-emissions tax makes energy storage more cost effective. Yong and McDonald[36] show that an emissions-tax regime that is set by a government with a willingness to commit to it, has a positive influence on the size and the direction of firm-level investment in clean technologies. Thus, adding a strong emissions tax to the already-established energy storage mandate in California may have beneficial economic, policy, and technology-development impacts. We also show that greater generator flexibility, which is

represented through a lower minimum-dispatchability requirement, reduces renewable curtailment and the amount of energy storage that is needed.

There are some important limitations of our analysis that can be examined in future research. The only environmental impact of electricity production and energy storage use that we examine is $CO_2$ emissions. There may be other important impacts. Our results show that PHS holds great promise, due to its relatively low cost. There are concerns around other environmental impacts of PHS, such as land and water use, species mortality, and impacts on biological production, however. Moreover, PHS is location-dependent and requires sites with specific characteristics[12]. The deployment of CAES is also limited, as specific underground formations are needed to store the compressed air[12]. Further examination of these limitations would provide a more comprehensive understanding of the deployment potential of these technologies.

Our optimization model could be applied to other case studies, with different generation mixes. We assume no degradation of

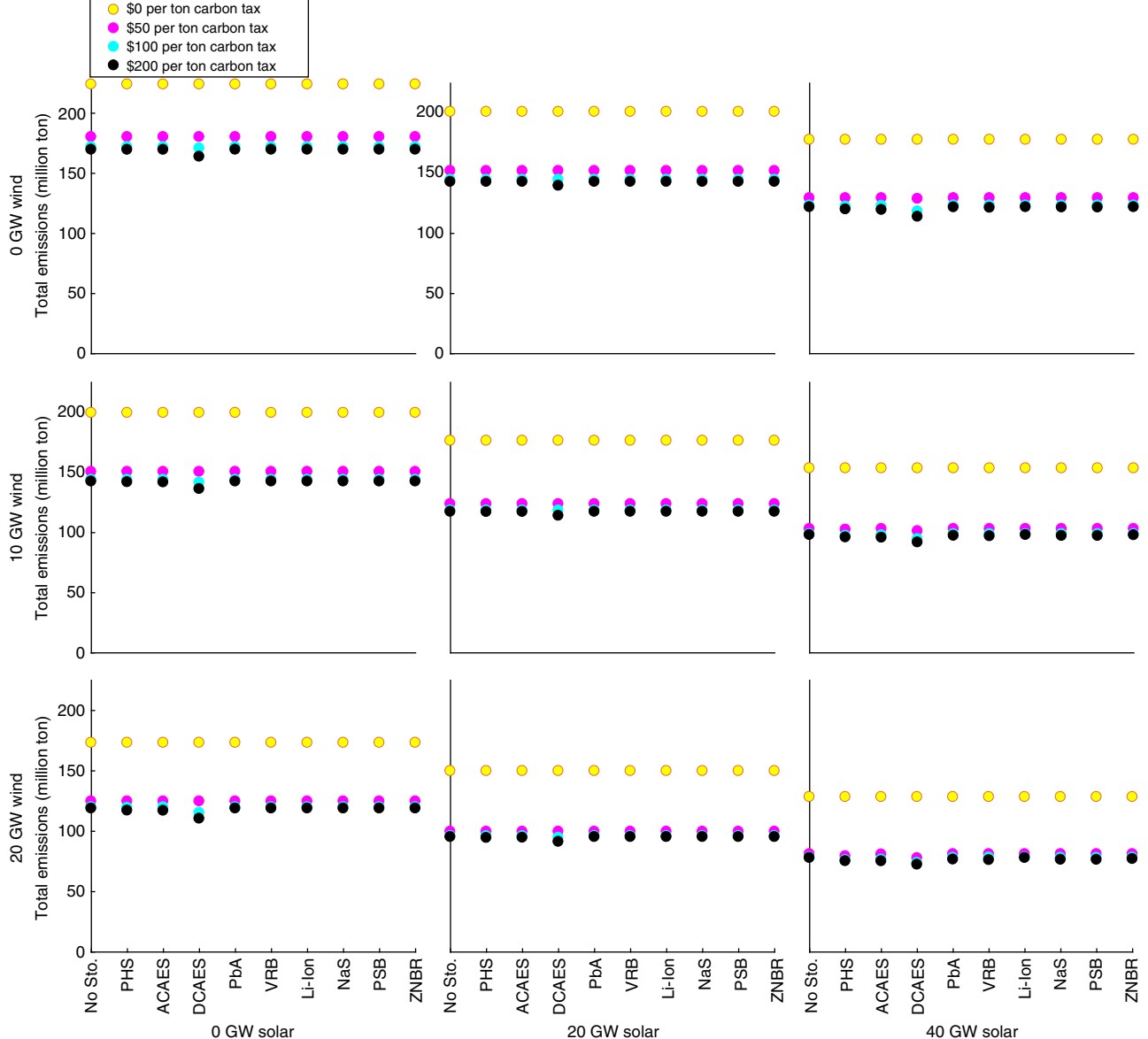

**Fig. 4** Annual $CO_2$ emissions (million ton) in Texas for the year 2012 in the base case with an 8.2-GW minimum-dispatchability requirement. Figure panels correspond to different wind- and solar-penetration levels, which are indicated at the left-hand side and bottom, respectively, of the figure. Source data are provided as a Source Data file

energy storage throughout its operation. Arbabzadeh et al.[37] show that its degradation does not change significantly the environmental impacts of using energy storage for generation-shifting. Nevertheless, future work could examine the impact of such degradation on the cost-effectiveness of using energy storage for alleviating renewable curtailment. We also assume that energy storage can operate between 0 and 100% state of charge. Future analyses can define technology-specific operational windows for energy storage.

## Methods

**Optimization model**. Our analysis uses an optimization model with an hourly time resolution over a $T$-h optimization horizon. The model determines the size of the energy storage system as well as the hourly operation of the power system. Specifically, we let $\bar{Q}$ and $\bar{S}$ denote the power and energy capacities of the energy storage, which are measured in MW and MWh, respectively. We let $g_{t,i}$ represent the hour-$t$ production level (measured in MW) of generator $i$, where $i \in \mathcal{I}$, the set of natural gas- and coal-fired, nuclear, biomass, hydroelectric, and geothermal generators. We let $\bar{R}_t$ and $R_t$ denote the total amount of renewable production that is available and the amount of renewable production that is used in hour $t$,

respectively. Both of these quantities are measured in MW. The difference, $(\bar{R}_t - R_t)$, gives hour-$t$ renewable curtailment. Finally, we let $q_t^c$ and $q_t^d$ denote MW that are charged into and discharged from energy storage, respectively, in hour $t$. We also let $s_t$ denote the ending hour-$t$ state of charge (SoC) of storage, which is measured in MWh.

The optimization model is formulated as:

$$\min_{\bar{Q}, \bar{S}, g, R, q^c, q^d, s} \kappa^Q \bar{Q} + \kappa^S \bar{S} + \sum_{t=1}^{T} \left[ (c^S + E\rho^S)q_t^d + \sum_{i \in \mathcal{I}} (c_i + E\rho_i)g_{t,i} \right] \quad (1)$$

$$\text{s.t.} \quad \sum_{i \in \mathcal{I}} g_{t,i} + R_t + q_t^d = L_t + q_t^c; \quad \forall t = 1, \dots, T; \quad (2)$$

$$\sum_{i \in \mathcal{I}} g_{t,i} + \min\{\bar{S}, s_{t-1}\} \geq \phi_t; \quad \forall t = 1, \dots, T; \quad (3)$$

$$0 \leq g_{t,i} \leq K_i; \quad \forall t = 1, \dots, T; i \in \mathcal{I}; \quad (4)$$

$$0 \leq R_t \leq \bar{R}_t; \quad \forall t = 1, \dots, T; \quad (5)$$

$$s_t = s_{t-1} + \eta^c q_t^c - q_t^d; \quad \forall t = 1, \dots, T; \quad (6)$$

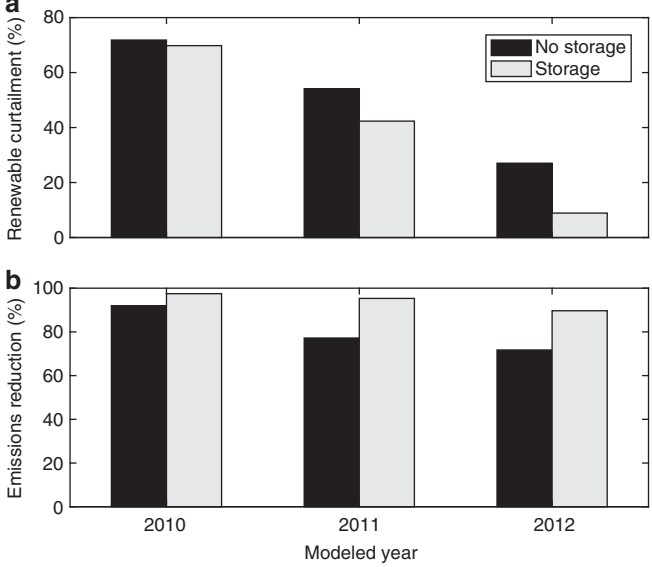

**Fig. 5** Annual renewable curtailment and $CO_2$-emissions reductions in California assuming 20-GW and 40-GW wind- and solar-penetration levels, respectively, $200 per ton $CO_2$-emissions tax, and base case 7.0-GW minimum-dispatchability requirement. Renewable curtailment reported as a percentage of potential renewable production and $CO_2$-emissions reductions are reported relative to a no-renewables case. Storage cases show results for **a** the technology that achieves the lowest curtailment and **b** highest emissions reductions in each year. Source data are provided as a Source Data file

$$0 \leq s_t \leq \bar{S}; \quad \forall t = 1, \ldots, T; \tag{7}$$

$$0 \leq q_t^c \leq \bar{Q}; \quad \forall t = 1, \ldots, T; \tag{8}$$

$$0 \leq q_t^d \leq \bar{Q}; \quad \forall t = 1, \ldots, T. \tag{9}$$

Eq. (1) minimizes the total cost of operating the system. The first two terms in the objective function, $\kappa^Q \bar{Q} + \kappa^S \bar{S}$, reflect the cost of building energy storage. Energy storage is assumed to have a capital cost that can depend on its power and energy capacities, with $\kappa^Q$ denoting the power-capacity cost (given in $ per MW) and $\kappa^S$ the energy-capacity cost (given in $ per MWh). The remaining term in the objective function denotes the hourly operating costs. Some energy storage technologies (e.g., DCAES) use a fuel, such as natural gas, when discharging stored energy. $c^S$ denotes the direct cost (in $ per MWh) of discharging stored energy for such technologies (i.e., $c^S = 0$ for technologies that do not consume fuel when discharging) while the term, $E \rho^S$, denotes any $CO_2$-related costs. Specifically, $E$ represents the $ per ton $CO_2$ tax and $\rho^S$ is the $CO_2$-emissions rate (in ton $MWh^{-1}$) of discharging stored energy. $c_i$ is the direct cost (in $ per MWh) of producing energy from generator $i$ and $\rho_i$ is the $CO_2$-emissions rate (in ton $MWh^{-1}$) of generator $i$.

Constraints (2) ensure that load is exactly met in each hour. We let $L_t$ denote the hour-$t$ load, in MW. Constraints (3) enforce the minimum-dispatchability requirement, where $\phi_t$ represents the hour-$t$ requirement in MW. If real-time renewable availability is sufficiently high, renewable generation is curtailed to ensure that the minimum-dispatchability requirement is met[4]. This dispatchability requirement can be met using generators (other than wind and solar), as well as energy storage. Constraints (4) and (5) impose generation limits on non-renewable and renewable generators, respectively. We let $K_i$ denote generator $i$'s production capacity in MW. Constraints (6) define the ending hour-$t$ SoC of energy storage to be the SoC at the end of hour $(t-1)$, plus any energy that is charged and less any energy that is discharged in hour $t$. We apply an efficiency factor, $\eta^c \in (0, 1)$, to the energy that is charged, which is a typical method of accounting for the round-trip efficiency losses of storing energy[38]. Constraints (7)–(9) impose the energy- and power-capacity constraints on the SoC and charging and discharging of energy storage, respectively.

The model is formulated using version 20170902 of the AMPL mathematical programming language and solved using version 12.7.1.0 of the CPLEX linear program solver.

**Annualization of capital cost of energy storage**. The capital costs of building each energy storage technology are annualized using a capital charge rate[39]. This annualization makes the capital costs comparable to the power system operating costs, which are modeled over a single-year period, in the optimization model. The capital charge rate takes into account the service life of each energy storage technology. In essence, a longer service life yields a lower capital charge rate, because the capital cost of building the energy storage can be amortized over a longer period. The capital charge rate, $\gamma$, is computed as:

$$\gamma = \delta + \frac{\delta}{(1+\delta)^Y - 1}, \tag{10}$$

where $Y$ is the service life of the technology and $\delta$ is the discount rate, which we take to be 10%. This yields capital charge rates ranging between 10% (for the technologies with 60-year service lives) and 16% (for ZNBR batteries, which have 10-year service lives).

**Wind and solar modeling**. The scenarios that we model vary the penetration of wind and photovoltaic solar exogenously. We consider cases with up to 20 GW and 40 GW of added wind and solar, respectively. The hourly generation that is available from the added wind plants are modeled using the Wind Integration National Dataset Toolkit (WIND Toolkit)[40]. The WIND Toolkit provides modeled historical wind-availability data for more than 126000 sites across the United States for the years 2007–2013. Because our other case study data are for the years 2010 through 2012, we use wind-availability data for the same years.

To capture the impacts of spatial diversification of the added wind, we compute hourly wind capacity factors that are averaged across each of the states of California and Texas. To do this, we let $\mathcal{W}$ denote the set of wind sites in each state that are in the WIND Toolkit. Then, we let $A_{t,w}$ denote the MW of wind that is available at location $w$ in hour $t$. We compute the state-average wind capacity factor in hour $t$, $W_t$, as:

$$W_t = \frac{\sum_{w \in \mathcal{W}} A_{t,w}}{\sum_{w \in \mathcal{W}} \bar{A}_w}, \tag{11}$$

where $\bar{A}_w$ is the assumed nameplate capacity of the wind generator at location $w$ in the WIND Toolkit dataset. The amount of wind that is available in hour-$t$ (in our optimization model) is computed as:

$$W_t \cdot \bar{W}, \tag{12}$$

where $\bar{W}$ is the aggregate amount of wind that is added to the system (i.e., $\bar{W}$ equals either 0 GW, 10 GW, or 20 GW).

Hourly solar availability is modeled in the same way, using modeled historical solar data that are obtained from the National Solar Radiation Database (NSRDB)[41,42]. The NSRDB data are processed using version 5 of the PVWatts software tool[43]. PVWatts simulates the output of a photovoltaic system, given solar and other weather-condition data. We assume that the added photovoltaic plants are fixed axis with a 180° azimuth and a tilt that is equal to each location's latitude. To account for geographic diversification of where solar can be added within each state, we compute state-average hourly solar capacity factors. To do this, we let $\mathcal{P}$ denote the set of 5636 and 1751 sites within the states of California and Texas, respectively, that are represented in the NSRDB dataset. Then, we define the state-average solar capacity factor in hour $t$, $P_t$, as:

$$P_t = \frac{\sum_{p \in \mathcal{P}} B_{t,p}}{\sum_{p \in \mathcal{P}} \bar{B}_p}, \tag{13}$$

where $B_{t,p}$ and $\bar{B}_p$ are the photovoltaic output that is available in hour $t$ and the assumed nameplate capacity of the photovoltaic generator at location $p$, respectively, in the NSRDB data. We then model available solar in hour $t$ (in our optimization model) as:

$$P_t \cdot \bar{P}, \tag{14}$$

where $\bar{P}$ is the aggregate amount of solar that is added to the system. We compute the total amount of renewable energy that is available in each hour as:

$$\bar{R}_t = W_t \cdot \bar{W} + P_t \cdot \bar{P}. \tag{15}$$

**Case studies—overview**. We examine using energy storage to ease the integration and reduce the curtailment of renewable energy in California and Texas. California makes for an interesting case study because it has limited ability to decarbonize through fuel switching (the fossil-fueled fleet in the state is almost entirely natural gas-fired). Concurrently, the state is pursuing ambitious renewable portfolio standards with the aim of decarbonization and is promoting more recently energy storage through policy measures. Given this context, Solomon et al.[44] evaluate the opportunities for increased use of renewable energy in California with and without energy storage. Eichman et al.[45] examine the value of California's energy storage mandate with high penetrations of renewable energy. They do not, however, endogenize the sizing of energy storage nor do they examine the range of technologies, renewable penetrations, and carbon-related policies that we do.

**Table 2 Installed generation capacity and annual-average load (MW)**

| | California | | | ERCOT | | |
|---|---|---|---|---|---|---|
| | **2010** | **2011** | **2012** | **2010** | **2011** | **2012** |
| Nuclear | 4577 | 4647 | 4647 | 4966 | 4960 | 4960 |
| Biomass | 1086 | 1156 | 1182 | 115 | 126 | 121 |
| Hydroelectric | 13850 | 13890 | 13901 | 689 | 689 | 689 |
| Geothermal | 2648 | 2648 | 2703 | 0 | 0 | 0 |
| Average Load | 20000 | 24430 | 26894 | 36335 | 38127 | 37017 |

Source: Form EIA-860 data from the United States Energy Information Administration

A second case study examines Texas, specifically focusing on the ERCOT system. ERCOT is largely electrically isolated from the rest of North America[46]. ERCOT makes for an interesting case relative to California, because it has greater variety in the mix of thermal generation, including coal- and natural gas-fired units, meaning that there is potential for fuel switching to achieve $CO_2$ reductions. Texas also has excellent renewable resources[46].

**Case studies—data**. Due to limited data availability, our case studies cover 20 April 2010 until 31 December 2012. During this period, California had about 237 natural gas-fired generating units whereas ERCOT had about 39 coal- and 234 natural gas-fired units installed. Table 2 summarizes the installed capacity of other generation technologies and the annual-average loads in the two systems.

The natural gas- and coal-fired generators are assumed to be dispatchable (i.e., their production levels can be varied to achieve supply/demand balance). The capacities and heat and $CO_2$-emissions rates of these generators are obtained from United States Environmental Protection Agency Air Markets Program Data (https://ampd.epa.gov/ampd/) and Form EIA-860 and EIA-923 data from the United States Energy Information Administration (EIA). The nuclear, biomass, hydroelectric, and geothermal generators (that were installed in the study years) are treated as being non-dispatchable. The outputs of these units are fixed based on historical hourly production levels that are reported by the CAISO and ERCOT. Table 3 summarizes the fuel prices that we use for the dispatchable natural gas- and coal-fired units, which are from Form EIA-923 data (these are reported in $ per MMBTU, as MMBTU is the unit that is used most commonly in the United States for reporting fuel prices). Fuel prices for other generating technologies are not needed, because these units are modeled as being non-dispatchable.

California exchanges energy with neighboring states. We assume these exchanges to be fixed in our case study, based on historical CAISO data. ERCOT has extremely limited energy exchanges, due to its being electrically isolated from the rest of North America. Thus, we model the ERCOT system as having no energy exchanges. CAISO and ERCOT also provide hourly historical load data, which we use in our analysis. We assume that the two systems have dispatchability requirements that the total output of the natural gas-fired, coal-fired, nuclear, biomass, hydroelectric, and geothermal generators plus the amount of energy that could be provided by energy storage be above some minimum value. This requirement reflects the limited flexibility of the non-renewable generators in reducing their output as well as the desire by system operators to maintain some dispatchable generation to accommodate unanticipated system contingencies[47]. We define the minimum-dispatchability requirement for the CAISO system based on an analysis of its flexibility[4] and generation and curtailment data that are published by CAISO. On this basis, we consider four different minimum-dispatchability requirements of 0.0 GW (i.e., the system is fully flexible with no minimum-dispatchability requirement), 5.4, 7.0, and 12.6 GW, with 7.0 GW as the base case. We set the minimum-dispatchability requirements for the ERCOT system by scaling on a pro rata basis compared to the values that are used for the CAISO system. This gives minimum-dispatchability requirements of 0.0, 6.3, 8.2, and 14.8 GW, with 8.2 GW as the base case, for the ERCOT system.

We examine nine energy storage technologies that have suitable characteristics for renewable integration- and curtailment-related applications: PHS, ACAES, DCAES, and PbA, VRB, Li-ion, NaS, PSB, and ZNBR batteries[8,37]. These technologies are characterized by their round-trip efficiencies, service lives, and capital costs, which are summarized in Table 4 and obtained from a comprehensive literature review[8,10–13,29,37,48–52]. The service lives of the technologies are accounted for when annualizing their capital costs. This annualization makes the capital costs of the technologies comparable to the cost of power system operations, which are modeled over a single-year period for each year that is studied. Because we only have data starting from 20 April 2010, we subannualize the capital cost in this year to make the capital and operating costs comparable.

The round-trip efficiency of DCAES is modeled differently than those of the other energy storage technologies. The other technologies are pure energy storage,

**Table 3 Fuel prices ($ per MMBTU)**

| Fuel | 2010 | 2011 | 2012 |
|---|---|---|---|
| Natural gas | 5.09 | 4.72 | 3.42 |
| Coal | 2.27 | 2.39 | 2.38 |

**Table 4 Technical characteristics of energy storage technologies**

| Technology | Round-trip efficiency | Service life (years) | Installation Cost | |
|---|---|---|---|---|
| | | | **Energy Component ($ per kWh)** | **Power Component ($ per kW)** |
| PHS[11–13,37,50] | 0.85 | 60 | 5 | 441 |
| ACAES[11–13,37] | 0.95 | 60 | 40 | 700 |
| DCAES[11–13,37] | a | 60 | 2 | 400 |
| PbA[10,12,13,37,50] | 0.90 | 15 | 200 | 222 |
| VRB[12,13,37,50] | 0.95 | 15 | 150 | 398 |
| Li-ion[12,37,48,51,52] | 0.90 | 15 | 320 | 620 |
| NaS[11–13,37,48,50,52] | 0.90 | 15 | 180 | 250 |
| PSB[13,50] | 0.85 | 15 | 120 | 330 |
| ZNBR[11,13,50] | 0.75 | 10 | 150 | 178 |

aThe efficiency of DCAES is modeled differently than those of other technologies. Each MWh of stored electricity can produce 1.39 MWh of electricity when discharged with 4.20 MMBTU of natural gas being consumed in the process[13,53]

in the sense that they each use electricity as the sole energy input. DCAES uses electricity when charging but combusts natural gas when discharged. Each MWh of electricity that is stored in a DCAES system is assumed to produce 1.39 MWh of electricity when discharged but uses 4.20 MMBTU of natural gas in doing so[13,53]. This natural gas combustion is assumed to result in $CO_2$ emissions of 0.058 ton $MMBTU^{-1}$ [37]. The direct cost of the natural gas that is consumed by the DCAES is computed using the values that are reported in Table 3.

**Case studies—scenarios**. For each energy storage technology, we model its optimal investment level and hourly operation of the power system in 36 scenarios that correspond to different renewable-penetration levels and carbon policies. These cases are examined in the CAISO and ERCOT systems for each of the years 2010–2012. Specifically, we examine three wind-penetration levels, which are cases with 0, 10, and 20 GW of wind total, three solar-penetration levels, with 0, 20, and 40 GW of photovoltaic solar total, and four carbon-tax regimes, with tax rates of $0, $50, $100, and $200 per ton. The outputs of the wind and solar plants can be curtailed, as required by the constraints of the optimization model to achieve hourly supply/demand balance.

## Data availability

The data that support the analysis within this paper and other findings of this study are available from the corresponding author upon reasonable request. The source data underlying Figs. 1–5 are provided as a Source Data file.

## Code availability

The optimization code that supports the analysis within this paper and other findings of this study are available from the corresponding author upon reasonable request.

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

## Acknowledgements

This work was supported by National Science Foundation Grants 1230236 and 1548015, the Dow Sustainability Fellows Program, and the University of Michigan Rackham Predoctoral Fellowship Program. We thank Y. Zerehsaz, H. Tavafoghi, and A. Sarabi (University of Michigan) for their valuable contributions on this project, the Center for Sustainable System at the University of Michigan for intellectual support, and A. Sorooshian (University of Arizona) for helpful discussions.

## Author contributions

M.A. and R.S. developed the model, designed the study, conducted the analysis, and co-wrote the paper. J.J. and G.K. participated in the study design and edited the paper.

## Additional information

**Competing interests:** The authors declare no competing interests.

