## [Peer Review File · Nature Communications]

Reviewer #1 (Remarks to the Author):

General comments

The paper provides some new insight but I am missing a distilled description of the significant new finding. What do the authors find to be most surprising and important in the results?

The authors use a case study (California in 2012) to explore the potential various storage technologies have to reduce curtailment of renewable energy and reduce overall greenhouse gas emissions. They design a series of scenarios with different renewable resource utilization of wind (0, 10, and 20 GW) and solar (0, 20, and 40 GW) and with various prices on carbon dioxide (0, 50, 100, and 200 \$/ton CO₂). For each of these scenarios, they determine optimal system operation, including how much storage capacity is installed of the different technologies. The authors then report for the various scenarios how much storage, and of which types, is installed and the storage capacities' (power and energy) impacts on renewable energy utilization and GHG reductions.

The introduction is a literature review that summarizes relevant papers. It is too long and contains unnecessary detail. Furthermore, these summaries are sometimes tangential, and overwhelm the argument(s) the authors try to make and framing they try to provide. I recommend shortening and focusing the introduction.

The case study is relevant due to the ambitious renewable portfolio standards and documented challenges presented by high penetration of solar generation combined with existing grid infrastructure. However, the authors only analyze one year and one state, limiting the broader applicability of their results.

The figures require significant improvements in clarity. In many cases, they could be replaced by tables. Figures contain too much unused (white) space. Fonts and symbols are too small. Captions contain typos, and subplots are labeled inconsistently from plot to plot.

In the discussion, there is no mention on how the introduction of carbon taxes, energy storage, and energy mixes, would impact electricity prices. What are the authors able to say about this?

Also in the discussion, the authors introduce their "reduced cost" analysis and provide results for only one technology (Li-ion). More detail on how this analysis was performed and its results for

other technologies should be included in the methods and results sections, or SI if deemed less relevant.

Their optimization model is extensively defined and explained in the methods section. The description is a bit too detailed to be included in the manuscript text; it is likely better suited for inclusion in the SI, while a simplified version should remain in the paper. However, their data and assumptions could use additional description and justification. This is especially important considering the disparate literature surrounding these estimates, that range from aspirational costs to real-world installed project estimates. The annualization of capital cost of energy storage also requires further clarification since it is not clear how the different technologies are compared while using different discount rates, different technology capacity decays, and SOC operation intervals.

In addition, a sensitivity analysis would significantly strengthen the results. For example, the authors mention that incorporation of lead-acid energy storage drops renewable curtailment from 12.4% to 11.9%. Is this decrease of 0.5% significant in light of the uncertainty of the underlying technology cost estimates? What is the impact of varying the dispatchability requirement?

Specific comments

The authors state “However, there are no studies in the extant literature that systematically investigate the economic viability of using energy storage to alleviate renewable curtailment for the purposes of decarbonizing electricity production.” but do not go on to fully address the issue themselves.

The authors should address possible bias resulting from the use of data from only 2012. Between 2011 and 2017, California experienced a prolonged drought, one of the worst recorded. How does the renewable resource availability in 2012 compare to availability in years not between 2011 and 2017? How does the variability in that availability compare? Similarly, how does electricity demand compare?

Throughout the manuscript, the authors sometimes mention greenhouse gases (GHGs) and in other places mention carbon dioxide (CO₂) emissions. In the data section, only “CO₂-emissions rates” are mentioned. Are the authors only exploring carbon dioxide emissions? Or are they considering other GHGs? If so, what emissions metric are they using? If the former, maybe change the language elsewhere (for example in the introduction). This is especially important considering the significant incorporation of various natural gas technologies, both in electricity generation and storage technologies, as methane and ethane emissions could be a significant contributors to the system’s overall GHG emissions (and targets). Similarly, what is the reference for the 0.058t/MBtu figure used in the modeling of DCAES? Is it also CO₂-only? Side note: In the conversion factor, considering that

the denominator is in US customary units, it would be useful to specify whether the “t” refers to metric or imperial tons. Alternatively, provide the value in t/MJ.

Both VRB and ZNBR technologies are estimated to have quite low energy costs for battery technologies, and a VRB efficiency of 0.95% is unexpectedly high. Could the authors provide more guidance as to which literature references influenced which estimates? In addition, some of the power estimates appear overly exact (222 \$/kW, 398 \$/kW as opposed to 220, 400). Is there a reason why?

The authors set their system’s dispatchability requirement at 12.6 GW, but other than citing a reference, do not discuss how this number was calculated or why it is appropriate for this study. More detail on how this number was determined would be useful. In addition, the sensitivity of their results to this value should be examined. Similarly, the wind, solar, and carbon tax values selection are not explained or supported with literature.

Statements such as, “Renewable curtailment is greatest in the spring, which is consistent with real-world experience in California.” should be supported by references.

In some places, the authors should clarify their language. In the methods section, when describing how wind and solar resources are incorporated into the model, the authors should avoid jargon, and maybe provide units to aid the reader. For example, the meaning of “hour-t wind” could be clarified. Another example: “in-state CO2” should probably read “in-state CO2 emissions.”

In Figure 2, the choice of PbA is not explained and the differences are barely perceptible. Possibly selecting a different set of storage technologies and showing the installed capacities (in a table) for the remaining technologies could improve clarity. Figure 2 could also be improved if installed capacities of storage were shown. Figures 3 and 4 would be improved if they also contained results for the ‘0 GW Wind’ and ‘0 GW Solar’ scenario to exhibit 2012 curtailment and emissions. Side note: Both figures and the table have inconsistent uses of capital and sentence case.

There is no discussion of the differences between nameplate and operation capacities (energy or power) for the different storage technologies. For example, Li-ion systems typically cannot use all of their capacity, from 0% SOC to 100% SOC.

Reviewer #2 (Remarks to the Author):

Referee Report for Nature Communications (NCOMMS-18-15867)

Title: The role of energy storage in deep decarbonization of electricity production in California

This paper analyzes the potential benefits of using energy storage technologies to reduce renewable energy curtailment and greenhouse gas (GHG) emissions under a carbon tax regime. By formulating a linear programming model and using the California electricity system in 2012 as a specific case, the authors showed that using energy storage technologies resulted in 93 percent greenhouse gas emissions (GHG) reductions compared to 2012 levels with 2 percent of renewable energy being curtailed. Without energy storage technologies, using similar renewable energy technologies only resulted in 76 percent greenhouse gas GHG reductions with close to 33 percent of renewable energy being curtailed.

Overall the paper is well-written and the result has clear practical and policy implications. I have a few remarks the authors might want to consider to improve the result and exposition.

(1) I wonder whether the authors could possibly conduct a similar analysis using data from other cities in US or from other locations in other countries as a form of stability analysis. Alternatively I wonder whether using the data from other years in California would lead to similar conclusion. This is just a robustness check but of course it depends whether there are data available.

(2) The overall structure of the paper ends with the optimization model. I would suggest for the model to be presented in the earlier section, probably before the section on data.

(3) The abstract should mention that the analysis include the consideration of a carbon tax.

(4) In the paper (page 3, end of paragraph 1), the authors mentioned that they assume different penetrations of solar and wind energy under different GHG policies, reflected by CO₂ taxes. I think this is somewhat confusing because the analysis is based on the same GHG policy (i.e., a carbon tax policy) but different degrees of policy intervention based on different carbon prices. I would suggest re-writing this.

Reviewer #3 (Remarks to the Author):

Overview

This paper investigates the role of energy storage in alleviating renewable generation curtailment experienced through the decarbonization process. Through a least-cost optimization model, the study demonstrates the optimal amount of energy storage (in terms of both power and energy capacity) given a number of scenarios in which the relative amounts of added wind and solar capacity are varied and a CO₂ tax is varied. The paper focuses on a case study of California because it is a state without the opportunity to switch from coal to natural gas to achieve significant carbon intensity reductions as most other states would have – this means California would have to look to renewables.

What are the major claims of the paper? Are the claims novel? If not, please identify the major papers that compromise novelty.

1. At current prices for energy storage systems, the value of energy storage in decarbonizing power systems results primarily from its ability to reduce renewable curtailment. The authors recognize that other services may demonstrate added value and suggest that this should be considered a lower bound on energy storage value

This claim is novel and is primary contribution of this paper. As the authors state in lines 42-43, there have been a number of papers that look at optimal sizing and operation of energy storage to achieve economic/environmental outcomes, the unique contribution here is exploring the question of how energy storage impacts curtailment with high penetration of wind and solar.

2. Capital cost of an energy storage technology is the primary factor in determining the optimal system size

3. CO₂ emissions taxes make energy storage more cost effective

These two claims are intuitive, and not as novel as the first claim. However, by providing them, it places confidence in the quality of the modeling work of the authors, and in that way adds value.

Will the paper be of interest to others in the field? Will the paper influence thinking in the field?

Yes – moving beyond papers that look at incremental increases in wind and solar penetration, this paper is more forward looking and would be likely to inform future modeling as well as decision and policy choices around the value of energy storage. Decarbonization and the integration of variable energy sources into the grid are highly important research areas, so this is quite topical.

Are the claims convincing? If not, what further evidence is needed?

Yes, the claims are supported by the experiments in terms of a California case study.

Are there other experiments that would strengthen the paper further? How much would they improve it, and how difficult are they likely to be?

Yes, while the case study in California is convincing (and of all the states, by far the best to begin with), it would be valuable to see if the trends are consistent in areas that are significantly different. In other words, do the case study findings generalize? The authors state on lines 322-323 that California is particularly unique. While results from CA are important because of the size of CA, the work may be enhanced by including another location or locations that can verify the generalizability of the findings. For example, comparing California to Texas which is also dominated by natural gas and could provide evidence for the generalizability of the findings. Comparing to Midwest ISO which is dominated by coal, represents a very different setting and may prove the point that is mentioned in lines 160-161 that if coal is an option, then a switch away from coal may be the dominant transition; additionally, if the results end up being similar to CA this would be very strong support for the generalizability of the studies findings.

While these are, admittedly likely be substantial analyses, even one of these efforts would help transition this work from a case study to a more generalizable study, which would likely increase its impact.

Are the claims appropriately discussed in the context of previous literature?

Generally speaking, yes they are. However, the claim on lines 42-43 regarding “A number of studies optimize the size and operation of energy storage...” should be supported with appropriate citations.

If the manuscript is unacceptable in its present form, does the study seem sufficiently promising that the authors should be encouraged to consider a resubmission in the future?

Yes – I think the area of research is worthwhile and as California is one of the first places in the country that will be facing large penetration levels of variable energy, this information is worthwhile for the larger community.

Technical comments:

A. For modeling energy storage, how are cycle lifetime and depth of discharge taken into account? Since the optimization algorithm is operating the storage system, different choices in operation will lead to different lifetimes for the devices.

B. On line 185, it's not entire clear why the dispatchability requirement is set to 12.6 GW – is there specific meaning for that number?

C. For modeling wind and solar, it sounds like the state average wind/solar hourly capacity factor is used. Will this lead to an overly-optimistic estimate of variability (i.e. the variability will be much less than the case if there are a number of discrete wind/solar farms scattered throughout the state)?

D. Will the code for this analysis be made available for reproducibility?

Organization comments:

E. The organization of the paper, ending with the Methods section, is unintuitive, and it would help readers to place the methods section before results.

Overall Recommendation: Revise and resubmit

Reviewer #1:

General comments

The paper provides some new insight but I am missing a distilled description of the significant new finding. What do the authors find to be most surprising and important in the results?

Thank you for taking the time to read and comment on our work. We appreciate the positive overall assessment and the keen observations and comments regarding areas for improvement.

This study provides a novel optimization study comparing the deployment of alternative energy storage technologies across renewable-integration levels to determine which energy storage technologies are most cost-effective in aiding renewable integration and reducing CO₂ emissions.

The results show that with relatively low emissions taxes (*i.e.*, \$50/t or less), PHES and CAES are the only economically viable technologies for averting renewable curtailment. Irrespective of the level of the carbon tax, energy storage is not cost-effective without added wind and solar capacity in California. This shows that the value of energy storage stems primarily from its ability to reduce renewable curtailment and displace fossil-fueled generation. In Texas, even in the zero renewable case, deploying some storage technologies such as CAES and PHES is cost-effective under higher emissions-tax regimes.

To sum up, our work has the following four critical takeaways.

- 1. Renewable curtailment is a key driver for cost effective energy storage deployment.**
- 2. Generator flexibility, which is represented through lower minimum dispatchable generation requirements, can reduce curtailment and the amount of energy storage that is needed.**
- 3. CO₂ taxes results in energy storage being more cost-effective for mitigating renewable curtailment.**
- 4. Lower installed cost of energy storage (as opposed to roundtrip efficiency of the energy-storage cycle) is a key driver for economically viable deployment.**

In light of this comment, the abstract is revised as follows:

“Decarbonizing electricity production is a challenge of growing importance. Deep decarbonization pathways likely require high penetrations of renewable energy (*e.g.*, solar and wind). A challenge with relying on renewable energy is that its real-time availability is weather dependent and may not be coincident with real-time electricity demand. One possible solution to this mismatch between renewable-availability and demand patterns is energy storage. Excess renewable energy that is available during periods of relatively low demand can be stored for use in subsequent periods of relatively low renewable availability. Without energy storage, excess renewable energy may otherwise be curtailed. In this paper we investigate the potential of currently available energy storage technologies to reduce renewable curtailment and CO₂ emissions in California and Texas under varying emissions tax regimes. We show that without energy storage, adding 20 GW of wind and 40 GW of solar to the California system achieves 72% CO₂ reductions (relative to a zero-renewable case) with close to one third of renewable energy being curtailed. Energy storage, on the other hand, allows 90% CO₂ reductions from the same renewable penetrations with as little as 9% of renewable energy being curtailed with some energy storage technologies. In Texas, the same renewable-deployment level leads to 54% emissions reductions with close to 3% of renewable energy being curtailed. Some energy storage technologies allow 57% emissions reductions with as little as 0.3% of renewable energy being curtailed. We also find that generator flexibility can reduce curtailment and the amount of energy storage that is needed for renewable integration.”

The authors use a case study (California in 2012) to explore the potential various storage technologies have to reduce curtailment of renewable energy and reduce overall greenhouse gas emissions. They design a series of scenarios with different renewable resource utilization of wind (0, 10, and 20 GW) and solar (0, 20, and 40 GW) and with various prices on carbon dioxide (0, 50, 100, and 200 \$/ton CO₂). For each of these scenarios, they determine optimal system operation, including how much storage capacity is installed of the different technologies. The authors then report for the various scenarios how much storage, and of which types, is installed and the storage capacities’ (power and energy) impacts on renewable energy utilization and GHG reductions.

The introduction is a literature review that summarizes relevant papers. It is too long and contains unnecessary detail. Furthermore, these summaries are sometimes tangential, and overwhelm the argument(s) the authors try to make and framing they try to provide. I recommend shortening and focusing the introduction.

Based on reviewer’s suggestion we have significantly revised and shortened the introduction.

The case study is relevant due to the ambitious renewable portfolio standards and documented challenges presented by high penetration of solar generation combined with existing grid infrastructure. However, the authors only analyze one year and one state, limiting the broader applicability of their results.

We appreciate this comment, as it does point to the limited scope of our initial analysis. To address this comment, we have expanded our case study to include the state of Texas as a second example. The following text has been added to the manuscript:

“As a second case study, we examine the state of Texas, specifically focusing on the Electric Reliability Council of Texas (ERCOT) system. ERCOT is largely electrically isolated from the rest of North America [39]. ERCOT also makes for an interesting case study relative to California, because it has greater variety in the mix of thermal generation, including coal- and natural gas-fired units, meaning that there is potential for fuel switching. The state of Texas also has excellent renewable resources [39].”

We have also expanded our analysis to include the years 2010 and 2011 in addition to 2012 for both California and Texas. The results of all of these cases are provided in the manuscript and Supplementary Information. The selection of years limited by data availability from the WIND Toolkit and NSRDB. We do observe similar trends among the years that are examined, but do point out some differences in the results between the years.

The figures require significant improvements in clarity. In many cases, they could be replaced by tables. Figures contain too much unused (white) space. Fonts and symbols are too small. Captions contain typos, and subplots are labeled inconsistently from plot to plot.

All figures are updated and revised to enhance clarity and incorporate our new results and Texas case study analysis. Fig.1 is replaced by Tables 4 and 5.

In the discussion, there is no mention on how the introduction of carbon taxes, energy storage, and energy mixes, would impact electricity prices. What are the authors able to say about this?

Also in the discussion, the authors introduce their “reduced cost” analysis and provide results for only one technology (Li-ion). More detail on how this analysis was performed and its results for other technologies should be included in the methods and results sections, or SI if deemed less relevant.

The concept of reduced cost comes from optimization theory (specifically, linear optimization theory). We provide a reference to the work of Sioshansi and Conejo [49], which provides a detailed derivation of how the reduced costs for a generic linear optimization problem are computed and how they are interpreted. Further, we provide reduced costs for other technologies and cases in the Supplementary Information. We have also added the following text to the manuscript:

“The reduced costs results for other storage technologies are provided in the Supplementary Information.”

Their optimization model is extensively defined and explained in the methods section. The description is a bit too detailed to be included in the manuscript text; it is likely better suited for inclusion in the SI,

while a simplified version should remain in the paper. However, their data and assumptions could use additional description and justification. This is especially important considering the disparate literature surrounding these estimates, that range from aspirational costs to real-world installed project estimates. The annualization of capital cost of energy storage also requires further clarification since it is not clear how the different technologies are compared while using different discount rates, different technology capacity decays, and SOC operation intervals.

We thank the reviewer for this comment. We believe that it is beneficial to keep the modeling methods in the paper, so that the paper can be largely self-contained. We have, instead, opted to relegate detailed results for the multitude of cases that we analyze to the Supplementary Information. However, if the editors feel that it is preferable to place the modeling methods in the Supplementary Information, we will happily do so.

For the costs of energy storage technologies, we have added sources from which our cost estimates are obtained (*cf.* Table 3).

In annualizing the capital costs of energy storage systems, we assume the same discount rate of 10%. This is explained in the manuscript by stating:

“where Y is the service life of the technology and δ is the discount rate, which we take to be 10%.”

For simplification, we have not assumed any capacity decay (or degradation) in our analysis. We recognize this as a limitation of our modeling, and say so in the discussion by stating that:

“We assume no degradation of energy storage throughout its operation. Arbabzadeh *et al.* [47] show that the annual degradation in the round-trip efficiency and capacity of energy storage do not significantly change the environmental impacts of using energy storage for generation-shifting. Nevertheless, a future study could examine the impact of such degradation on the cost-effectiveness of using energy storage for alleviating renewable curtailment.”

We further added the following text:

“We also assume that energy storage can operate between 0% and 100% state of charge. Future analyses can define various operational windows for different energy storage technologies that are technology-specific.”

to the discussion, which notes another limitation of our analysis which could be a topic of future study.

In addition, a sensitivity analysis would significantly strengthen the results. For example, the authors mention the that incorporation of lead-acid energy storage drops renewable curtailment from 12.4% to 11.9%. Is this decrease of 0.5% significant in light of the uncertainty of the underlying technology cost estimates? What is the impact of varying the dispatchability requirement?

To address this and another review comment, Fig. 2 and the associated text regarding lead-acid batteries has been removed from the manuscript.

To analyze the sensitivity of the results to the capital costs of energy storage technologies, we have examined additional cases for Li-ion, NaS, and PSB batteries in which the costs of these technologies are changed relative to baseline values. We have added the following text:

“Given the wide range of costs for Li-ion, NaS, and PSB batteries that are reported in the literature, we also conduct a sensitivity analysis, in which the capital costs of Li-ion batteries are reduced to \$259/kWh and \$59/kW, the costs of NaS batteries are increased to \$350/kW and \$350/kWh, and the costs of PSB batteries are reduced to \$200/kW and \$90/kWh. Table 6 summarizes the impacts of these changed capital costs on renewable curtailments and CO₂ emissions. Specifically, the table reports changes in renewable curtailments and CO₂ emissions relative to the levels that are achieved with the baseline costs that are in Table 3, as a percentage of the base-line curtailment and emissions impacts of Li-ion, NaS, and PSB. The results that are in the table are for 2012, assuming 20 GW of wind and 40 GW of solar are added to each system, a \$200/t CO₂-emissions tax, and the base-case minimum-dispatchability requirement for each system. The amounts of energy storage added, renewable curtailments, and CO₂ emissions that are achieved in other scenarios are provided in the Supplementary Information.”

to the discussion section, which introduces this sensitivity analysis.

To address the comment regarding the dispatchability requirement, we now examine multiple dispatchable requirements in our analysis. The manuscript has been revised to state that:

“We assume that the two systems have dispatchability requirements that the total output of the natural gas-fired, coal-fired, nuclear, biomass, hydroelectric, and geothermal generators plus the amount of energy that could be provided by the energy storage system be above some minimum value. This requirement reflects the limited flexibility of the non-renewable generators in reducing their output as well as the desire by system operators to maintain some dispatchable generation to accommodate unanticipated system contingencies [40]. For the CAISO system, we define the minimum-dispatchability requirement based on an analysis of the system’s flexibility [7] and generation and curtailment data that are published by CAISO. Based on these data sources, we consider four different minimum-dispatchability requirements of 0.0 GW (meaning that the system is fully flexible and has no minimum-dispatchability requirement), 5.4 GW, 7.0 GW, and 12.6 GW, with 7.0 GW as the base case. We set the minimum-dispatchability requirements for the ERCOT system by scaling on a *pro rata* basis compared to the values that are used for the CAISO system. This gives minimum-dispatchability requirements of 0.0 GW, 6.3 GW, 8.2 GW, and 14.8 GW, with 8.2 GW as the base case, for the ERCOT system.”

The authors state “However, there are no studies in the extant literature that systematically investigate the economic viability of using energy storage to alleviate renewable curtailment for the purposes of decarbonizing electricity production.” but do not go on to fully address the issue themselves.

We appreciate this comment, but respectfully disagree. Other works in the existing literature do not examine the question of what role energy storage plays in reducing renewable curtailment and aiding the decarbonization of electricity production as we do. Rather, most other works examine an

exogenously fixed amount of energy storage and examine its impacts with renewables. What we do is to systematically examine nine currently available energy storage technologies that are differentiated by their capital costs, round-trip efficiencies, and service lives. Our model allows us to determine the exact amount of energy storage capacity that would be built to most cost effectively reduce the cost of operating a system by reducing renewable curtailment. We also examine how emissions taxes and the operating flexibility of the underlying power system impact this potential use of energy storage. By examining the reduced costs that are derived from solving our linear optimization model, we are also able to determine the extent to which the cost of a currently uneconomic energy storage technology must come down to make it cost-effective for this proposed use. As such, we argue that we provide a methodical analysis of the role of energy storage in helping to decarbonize electricity production through the use of variable renewable energy.

The authors should address possible bias resulting from the use of data from only 2012. Between 2011 and 2017, California experienced a prolonged drought, one of the worst recorded. How does the renewable resource availability in 2012 compare to availability in years not between 2011 and 2017? How does the variability in that availability compare? Similarly, how does electricity demand compare?

We have expanded our analysis to include year 2010, which is not between 2011 and 2017. By estimating the solar and wind capacity factors in 2010 we are capturing the renewable variability in that year compared to 2011 and 2012. Table 1 summarizes generation capacities of non-dispatchable resources as well as average demand in California and Texas in years 2010, 2011, and 2012. The table shows that there were relatively small changes in the availability of these non-dispatchable resources among these three years in both systems. However, there was significant load growth in California between 2010 and 2012, which does give rise to some differences in renewable-curtailment rates.

Table 1. Installed generation capacity and annual-average load (MW).

	California			ERCOT		
	2010	2011	2012	2010	2011	2012
Nuclear	4577	4647	4647	4966	4960	4960
Biomass	1086	1156	1182	115	126	121
Hydroelectric	13850	13890	13901	689	689	689
Geothermal	2648	2648	2703	0	0	0
Average Load	20000	24430	26894	36335	38127	37017

Throughout the manuscript, the authors sometimes mention greenhouse gases (GHGs) and in other places mention carbon dioxide (CO2) emissions. In the data section, only “CO2-emissions rates” are mentioned. Are the authors only exploring carbon dioxide emissions? Or are they considering other GHGs? If so, what emissions metric are they using? If the former, maybe change the language elsewhere (for example

in the introduction). This is especially important considering the significant incorporation of various natural gas technologies, both in electricity generation and storage technologies, as methane and ethane emissions could be a significant contributors to the system's overall GHG emissions (and targets). Similarly, what is the reference for the 0.058t/MBtu figure used in the modeling of DCAES? Is it also CO₂-only? Side note: In the conversion factor, considering that the denominator is in US customary units, it would be useful to specify whether the “t” refers to metric or imperial tons. Alternatively, provide the value in t/MJ.

Thank you for pointing this out. We are only exploring CO₂ emissions. We have edited the manuscript to replace “GHG emissions” with “CO₂ emissions” throughout. Reference [41] is also added as the source of for the 0.058 ton/MMBtu associated with the DCAES plant. We have added the following text:

“We further assume that this combustion of natural gas results in CO₂ emissions of 0.058 ton/MMBtu [41].”

clarifying this source. In this manuscript “t” refers to short tons. We have edited the manuscript to replace “t” with “ton” throughout, to avoid any ambiguity.

Both VRB and ZNBR technologies are estimated to have quite low energy costs for battery technologies, and a VRB efficiency of 0.95% is unexpectedly high. Could the authors provide more guidance as to which literature references influenced which estimates? In addition, some of the power estimates appear overly exact (222 \$/kW, 398 \$/kW as opposed to 220, 400). Is there a reason why?

We have modified Table 3 to include the references from which the costs, efficiencies, and other characteristics are obtained. The costs are taken directly from these references, with no particular reason that some of them are overly exact.

The authors set their system's dispatchability requirement at 12.6 GW, but other than citing a reference, do not discuss how this number was calculated or why it is appropriate for this study. More detail on how this number was determined would be useful. In addition, the sensitivity of their results to this value should be examined. Similarly, the wind, solar, and carbon tax values selection are not explained or supported with literature.

We appreciate this comment. To address this, we conducted additional sensitivity analyses in which the dispatchability requirement was varied. We have added the following text:

“We assume that the two systems have dispatchability requirements that the total output of the natural gas-fired, coal-fired, nuclear, biomass, hydroelectric, and geothermal generators plus the amount of energy that could be provided by the energy storage system be above some minimum value. This requirement reflects the limited flexibility of the non-renewable generators in reducing their output as well as the desire by system operators to maintain some dispatchable generation to accommodate unanticipated system contingencies [40]. For the CAISO system, we define the

minimum-dispatchability requirement based on an analysis of the system’s flexibility [7] and generation and curtailment data that are published by CAISO. Based on these data sources, we consider four different minimum-dispatchability requirements of 0.0 GW (meaning that the system is fully flexible and has no minimum-dispatchability requirement), 5.4 GW, 7.0 GW, and 12.6 GW, with 7.0 GW as the base case. We set the minimum-dispatchability requirements for the ERCOT system by scaling on a *pro rata* basis compared to the values that are used for the CAISO system. This gives minimum-dispatchability requirements of 0.0 GW, 6.3 GW, 8.2 GW, and 14.8 GW, with 8.2 GW as the base case, for the ERCOT system.”

which discusses these sensitivity analyses.

Statements such as, “Renewable curtailment is greatest in the spring, which is consistent with real-world experience in California.” should be supported by references.

We have added a reference to the work of Denholm *et al.* [7], which discusses the fact that renewable curtailment tends to be greater during the spring as opposed to other seasons in California.

In some places, the authors should clarify their language. In the methods section, when describing how wind and solar resources are incorporated into the model, the authors should avoid jargon, and maybe provide units to aid the reader. For example, the meaning of “hour-*t* wind” could be clarified. Another example: “in-state CO₂” should probably read “in-state CO₂ emissions.”

We have modified the “Methods” section of the manuscript by adding the text:

“To capture the impacts of spatial diversification of the added wind, we compute hourly wind capacity factors that are averaged across each of the states of California and Texas. To do this, we let \mathcal{W} denote the set of wind sites in each state that are in the WIND Toolkit. We then let $A_{t,w}$ denote the MW wind that is available at location w in hour t . We compute the state-average wind capacity factor at hour t , \mathcal{W}_t as:

$$\mathcal{W}_t = \frac{\sum_{w \in \mathcal{W}} A_{t,w}}{\sum_{w \in \mathcal{W}} \bar{A}_w} \quad (11)”$$

to better describe how hour-*t* wind is computed.

The phrase “in-state CO₂” has been replaced by “in-state CO₂ emissions” in the manuscript.

In Figure 2, the choice of PbA is not explained and the differences are barely perceptible. Possibly selecting a different set of storage technologies and showing the installed capacities (in a table) for the remaining technologies could improve clarity. Figure 2 could also be improved if installed capacities of storage were shown. Figures 3 and 4 would be improved if they also contained results for the ‘0 GW

Wind' and '0 GW Solar' scenario to exhibit 2012 curtailment and emissions. Side note: Both figures and the table have inconsistent uses of capital and sentence case.

On the basis of this and other comments, we have removed Fig. 2. We have added the '0 GW Wind' and '0 GW Solar' scenario to Figs. 3 and 4, which show annual CO₂ emissions.

There is no discussion of the differences between nameplate and operation capacities (energy or power) for the different storage technologies. For example, Li-ion systems typically cannot use all of their capacity, from 0% SOC to 100% SOC.

Thank you for this comment. We make the simplifying assumption that all energy storage technologies can operate between the 0% to 100% SOC range. However, this is an important *caveat* of our results. We have added the following text:

“We also assume that energy storage can operate between 0% and 100% state of charge. Future analyses can define various operational windows for different energy storage technologies that are technology-specific.”

which notes this assumption and lists it as a possible area for future research.

Reviewer #2:

This paper analyzes the potential benefits of using energy storage technologies to reduce renewable energy curtailment and greenhouse gas (GHG) emissions under a carbon tax regime. By formulating a linear programming model and using the California electricity system in 2012 as a specific case, the authors showed that using energy storage technologies resulted in 93 percent greenhouse gas emissions (GHG) reductions compared to 2012 levels with 2 percent of renewable energy being curtailed. Without energy storage technologies, using similar renewable energy technologies only resulted in 76 percent greenhouse gas GHG reductions with close to 33 percent of renewable energy being curtailed.

Overall the paper is well-written and the result has clear practical and policy implications. I have a few remarks the authors might want to consider to improve the result and exposition.

Thank you for taking the time to review and comment on our paper and for the positive overall assessment of our work.

(1) I wonder whether the authors could possibly conduct a similar analysis using data from other cities in US or from other locations in other countries as a form of stability analysis. Alternatively I wonder whether using the data from other years in California would lead to similar conclusion. This is just a robustness check but of course it depends whether there are data available.

We appreciate this comment, as it does point to the limited scope of our initial analysis. To address this comment, we have expanded our case study to include the state of Texas as a second example. The following text has been added to the manuscript:

“As a second case study, we examine the state of Texas, specifically focusing on the Electric Reliability Council of Texas (ERCOT) system. ERCOT is largely electrically isolated from the rest of North America [39]. ERCOT also makes for an interesting case study relative to California, because it has greater variety in the mix of thermal generation, including coal- and natural gas-fired units, meaning that there is potential for fuel switching. The state of Texas also has excellent renewable resources [39].”

We have also expanded our analysis to include the years 2010 and 2011 in addition to 2012 for both California and Texas. The results of all of these cases are provided in the manuscript and Supplementary Information. The selection of years limited by data availability from the WIND Toolkit and NSRDB. We do observe similar trends among the years that are examined, but do point out some differences in the results between the years.

(2) The overall structure of the paper ends with the optimization model. I would suggest for the model to be presented in the earlier section, probably before the section on data.

We appreciate this comment. Normally, we would completely agree that it is preferable to present a generic model before a specific case study. However, the house style of Nature Communications has the “Methods” section presented at the end of the article. We have opted for this structure to fit the

standard style of the journal. We should also note that other reviewers suggested relegating the model to the Supplementary Information. We believe that placing the model in the Methods section at the end of the manuscript is a fair compromise between these conflicting views.

(3) The abstract should mention that the analysis include the consideration of a carbon tax.

Thank you for noticing this and points it out. The abstract is revised to read:

“In this paper we investigate the potential of currently available energy storage technologies to reduce renewable curtailment and CO₂ emissions in California and Texas under varying emissions tax regimes.”

which more clearly makes this point.

(4) In the paper (page 3, end of paragraph 1), the authors mentioned that they assume different penetrations of solar and wind energy under different GHG policies, reflected by CO₂ taxes. I think this is somewhat confusing because the analysis is based on the same GHG policy (i.e., a carbon tax policy) but different degrees of policy intervention based on different carbon prices. I would suggest re-writing this.

We appreciate this comment. The sentence referencing the cases that are considered has been modified to read:

“We assume varying penetrations of solar and wind, under different CO₂-emissions policies, which are reflected by CO₂ taxes.”

which should be more clear.

Reviewer #3:

Overview

This paper investigates the role of energy storage in alleviating renewable generation curtailment experienced through the decarbonization process. Through a least-cost optimization model, the study demonstrates the optimal amount of energy storage (in terms of both power and energy capacity) given a number of scenarios in which the relative amounts of added wind and solar capacity are varied and a CO₂ tax is varied. The paper focuses on a case study of California because it is a state without the opportunity to switch from coal to natural gas to achieve significant carbon intensity reductions as most other states would have – this means California would have to look to renewables.

What are the major claims of the paper? Are the claims novel? If not, please identify the major papers that compromise novelty.

1. At current prices for energy storage systems, the value of energy storage in decarbonizing power systems results primarily from its ability to reduce renewable curtailment. The authors recognize that other services may demonstrate added value and suggest that this should be considered a lower bound on energy storage value

This claim is novel and is primary contribution of this paper. As the authors state in lines 42-43, there have been a number of papers that look at optimal sizing and operation of energy storage to achieve economic/environmental outcomes, the unique contribution here is exploring the question of how energy storage impacts curtailment with high penetration of wind and solar.

2. Capital cost of an energy storage technology is the primary factor in determining the optimal system size

3. CO₂ emissions taxes make energy storage more cost effective

These two claims are intuitive, and not as novel as the first claim. However, by providing them, it places confidence in the quality of the modeling work of the authors, and in that way adds value.

Thank you for taking the time to review and comment on our work. We also appreciate the positive overall assessment.

We agree with respect to the major takeaways from our analysis. To sum up, our work has the following four critical takeaways.

- 1. Renewable curtailment is a key driver for cost effective energy storage deployment.**
- 2. Generator flexibility, which is represented through lower minimum dispatchable generation requirements, can reduce curtailment and the amount of energy storage that is needed.**
- 3. CO₂ taxes results in energy storage being more cost-effective for mitigating renewable curtailment.**

4. Lower installed cost of energy storage (as opposed to roundtrip efficiency of the energy-storage cycle) is a key driver for economically viable deployment.

Based on other review comments, we have also added a case study involving the state of Texas. Texas provides a nice contrasting case to California, exactly because it *does* have the opportunity for decarbonization through fuel switching (which the reviewer notes California does not have).

Will the paper be of interest to others in the field? Will the paper influence thinking in the field?

Yes – moving beyond papers that look at incremental increases in wind and solar penetration, this paper is more forward looking and would be likely to inform future modeling as well as decision and policy choices around the value of energy storage. Decarbonization and the integration of variable energy sources into the grid are highly important research areas, so this is quite topical.

Are the claims convincing? If not, what further evidence is needed?

Yes, the claims are supported by the experiments in terms of a California case study.

Are there other experiments that would strengthen the paper further? How much would they improve it, and how difficult are they likely to be?

Yes, while the case study in California is convincing (and of all the states, by far the best to begin with), it would be valuable to see if the trends are consistent in areas that are significantly different. In other words, do the case study findings generalize? The authors state on lines 322-323 that California is particularly unique. While results from CA are important because of the size of CA, the work may be enhanced by including another location or locations that can verify the generalizability of the findings. For example, comparing California to Texas which is also dominated by natural gas and could provide evidence for the generalizability of the findings. Comparing to Midwest ISO which is dominated by coal, represents a very different setting and may prove the point that is mentioned in lines 160-161 that if coal is an option, then a switch away from coal may be the dominant transition; additionally, if the results end up being similar to CA this would be very strong support for the generalizability of the studies findings.

While these are, admittedly likely be substantial analyses, even one of these efforts would help transition this work from a case study to a more generalizable study, which would likely increase its impact.

We appreciate this comment, as it does point to the benefit of expanding the analysis to consider a system with a fundamentally different underlying generation mix. To address this comment, we have expanded our case study to include the state of Texas as a second example. The following text has been added to the manuscript:

“As a second case study, we examine the state of Texas, specifically focusing on the Electric Reliability Council of Texas (ERCOT) system. ERCOT is largely electrically isolated from the rest of North America [39]. ERCOT also makes for an interesting case study relative to California, because it has greater variety in the mix of thermal generation, including coal- and natural gas-fired units, meaning that there is potential for fuel switching. The state of Texas also has excellent renewable resources [39].”

Are the claims appropriately discussed in the context of previous literature?

Generally speaking, yes they are. However, the claim on lines 42-43 regarding “A number of studies optimize the size and operation of energy storage...” should be supported with appropriate citations.

Thank you for this suggestion. We have added the following text:

“In addition to economic considerations, a number of studies model emissions in optimizing storage use. Hemmati *et al.* [34] develop a multistage generation expansion planning model that shows that adding energy storage may decrease the total costs and emissions, due to the reduced need for building peaking generation capacity. de Sisternes *et al.* [35] model the Texas electric grid to determine the optimal mix of generation capacity and energy storage in 2035 at minimum cost. They show that when subjected to CO₂ limits, the role of energy storage increases as the limits become more stringent. This is because energy storage allows more efficient use of low-cost/low-CO₂ generation resources. Their analysis treats the capacity of two generic energy storage technologies as being exogenously fixed. Thus, they do not consider specific technologies nor do they fully optimize the size of the storage technologies endogenously. Arciniegas and Hittinger [36] optimize the operation and location of energy storage to maximize revenue and reduce CO₂ emissions within a number of subregions of the United States. Their results show that adding CO₂ emissions to the objective function of the optimization can yield large reductions in energy storage-related emissions at minimal expense to the owner of the energy storage system.”

after lines 42-43 summarizing relevant optimization-based works in the extant literature.

If the manuscript is unacceptable in its present form, does the study seem sufficiently promising that the authors should be encouraged to consider a resubmission in the future?

Yes – I think the area of research is worthwhile and as California is one of the first places in the country that will be facing large penetration levels of variable energy, this information is worthwhile for the larger community.

Technical comments:

A. For modeling energy storage, how are cycle lifetime and depth of discharge taken into account? Since the optimization algorithm is operating the storage system, different choices in operation will lead to different lifetimes for the devices.

Energy storage devices are assumed to have a fixed service life, which is technology dependent. Table 3 summarizes the assumed service lives of the technologies, and provides sources from which these data are obtained.

B. On line 185, it's not entirely clear why the dispatchability requirement is set to 12.6 GW – is there specific meaning for that number?

We appreciate this comment. The dispatchability requirement is based on an NREL study of California. However, as other reviewers have pointed out, a sensitivity analysis to determine the extent to which our results are driven by this assumption would be beneficial. As such, we include cases in which this parameter is varied for both the California and ERCOT systems. We have added the following text:

“We assume that the two systems have dispatchability requirements that the total output of the natural gas-fired, coal-fired, nuclear, biomass, hydroelectric, and geothermal generators plus the amount of energy that could be provided by the energy storage system be above some minimum value. This requirement reflects the limited flexibility of the non-renewable generators in reducing their output as well as the desire by system operators to maintain some dispatchable generation to accommodate unanticipated system contingencies [40]. For the CAISO system, we define the minimum-dispatchability requirement based on an analysis of the system’s flexibility [7] and generation and curtailment data that are published by CAISO. Based on these data sources, we consider four different minimum-dispatchability requirements of 0.0 GW (meaning that the system is fully flexible and has no minimum-dispatchability requirement), 5.4 GW, 7.0 GW, and 12.6 GW, with 7.0 GW as the base case. We set the minimum-dispatchability requirements for the ERCOT system by scaling on a *pro rata* basis compared to the values that are used for the CAISO system. This gives minimum-dispatchability requirements of 0.0 GW, 6.3 GW, 8.2 GW, and 14.8 GW, with 8.2 GW as the base case, for the ERCOT system.”

to the manuscript introducing these sensitivity cases.

C. For modeling wind and solar, it sounds like the state average wind/solar hourly capacity factor is used. Will this lead to an overly-optimistic estimate of variability (i.e. the variability will be much less than the case if there are a number of discrete wind/solar farms scattered throughout the state)?

It could, to some extent. However, we believe that a state-average capacity factor will be much more reflective of actual wind and solar profiles that would be observed with the high penetrations of wind and solar that we model. This is simply because at the tens of GW scale, wind and solar plants must, by definition, be geographically dispersed. Thus, we believe that our modeling approach is preferable to using a small subset of locations.

D. Will the code for this analysis be made available for reproducibility?

Yes, it all codes and input data will be available upon reasonable request.

Organization comments:

E. The organization of the paper, ending with the Methods section, is unintuitive, and it would help readers to place the methods section before results.

We would normally completely agree with this. However, we have opted to structure the paper in this way based on the standard house style that is employed by Nature Communications. If the editors direct us to move the modeling into the body of the manuscript, we will gladly do so.

Overall Recommendation: Revise and resubmit

Thank you again for the positive overall assessment of our work.

Reviewers' comments:

Reviewer #2 (Remarks to the Author):

Thank you for the effort to address my remarks and suggestions. I have no other comments.

Reviewer #3 (Remarks to the Author):

This manuscript with the addition of California as an added case study has been greatly improved in terms of the generalizability of the findings, so I thank the authors for their conscientiousness in addressing reviewer comments. I think the content from a technical perspective is very good, and the methodology sound. Also, descriptions of some of the takeaways is improved stating the novelty of the work more clearly.

I think the minor changes that remain to optimize this manuscript are in presentation and conciseness, and that's what the remainder of this review focuses on. Parts of the results section and discussion section, now that the paper has been expanded, could be made more concise and to clearly provide the supporting evidence for the author's four key takeaways that they outline in their response to reviewers.

Tables 4 and 5

Tables 4 and 5 should be moved into the supplemental information as they're too sparse for inclusion in the body of the manuscript, although the information contained within them is quite valuable.

Figures 1-4

Figures 1 and 2 are challenging for extracting meaningful insights from given the numbers of overlapping symbols and small text of the x-axis. Also, the x-axis contains significant redundant text (namely the words GW, Wind, and Solar). You could have xlabel labels structured in a more concise and easy to read way, for example:

Wind (GW):	0	0	0	10	10	10	...
Solar (GW):	0	20	40	0	20	40	...

Figure 1 and 2 need to be revised to more clearly support your takeaway points, otherwise, the excellent analysis and justification for your findings will be lost on most readers.

Also, consider making the y-axes the same for all of these plots to facilitate comparison.

Also, Figures 1-4 are hard to tell one from the other. I'd recommend making it clear what distinguishes each figure. It took me a few glances before I realized one set was renewable curtailment and the other for CO2 emissions.

For Figures 3 and 4, you may want to position these as changes in emissions over the baseline (this could be done as a fraction of the total or per-MWh)

Discussion

Overall, the discussion section is also improved. I think that the four key points that the authors identified in their response are conveyed, although this could be made clearer by making four paragraphs in the discussion clearly map to those four points. While this is broadly done now, leading with the key takeaway in each paragraph would make the value of this paper more explicit.

It may be worthwhile to convey in the discussion some of the lessons learned in lines 245-261 about how the specific fuel mixes in CAISO and ERCOT. Explaining how this analysis shows two ends of a spectrum that most ISOs will fall somewhere in between will add weight to the findings.

Reviewer #3 (Remarks to the Author):

This manuscript with the addition of California as an added case study has been greatly improved in terms of the generalizability of the findings, so I thank the authors for their conscientiousness in addressing reviewer comments. I think the content from a technical perspective is very good, and the methodology sound. Also, descriptions of some of the takeaways is improved stating the novelty of the work more clearly.

I think the minor changes that remain to optimize this manuscript are in presentation and conciseness, and that's what the remainder of this review focuses on. Parts of the results section and discussion section, now that the paper has been expanded, could be made more concise and to clearly provide the supporting evidence for the author's four key takeaways that they outline in their response to reviewers.

Tables 4 and 5

Tables 4 and 5 should be moved into the supplemental information as they're too sparse for inclusion in the body of the manuscript, although the information contained within them is quite valuable.

***Nature Communications* discourages placing results as Supplementary Information and instead encourages all tables and results to be in the body of the manuscript. Despite this, we agree with the comment that tables are too sparse to have in the body and have made the suggested change.**

Figures 1-4

Figures 1 and 2 are challenging for extracting meaningful insights from given the numbers of overlapping symbols and small text of the x-axis. Also, the x-axis contains significant redundant text (namely the words GW, Wind, and Solar). You could have xlabel's structured in a more concise and easy to read way, for example:

Wind (GW): 0 0 0 10 10 10 ...
Solar (GW): 0 20 40 0 20 40 ...

Figure 1 and 2 need to be revised to more clearly support your takeaway points, otherwise, the excellent analysis and justification for your findings will be lost on most readers. Also, consider making the y-axes the same for all of these plots to facilitate comparison. Also, Figures 1-4 are hard to tell one from the other. I'd recommend making it clear what distinguishes each figure. It took me a few glances before I realized one set was renewable curtailment and the other for CO₂ emissions.

For Figures 3 and 4, you may want to position these as changes in emissions over the baseline (this could be done as a fraction of the total or per-MWh)

Figures 1-4 are revised based on reviewer's comments.

Discussion

Overall, the discussion section is also improved. I think that the four key points that the authors identified in their response are conveyed, although this could be made clearer by making four paragraphs in the discussion clearly map to those four points. While this is broadly done now, leading with the key takeaway in each paragraph would make the value of this paper more explicit.

It may be worthwhile to convey in the discussion some of the lessons learned in lines 245-261 about how the specific fuel mixes in CAISO and ERCOT. Explaining how this analysis shows two ends of a spectrum that most ISOs will fall somewhere in between will add weight to the findings.

The discussion is revised as follows:

Our case study shows that energy storage can play a non-trivial role in decarbonizing California's electricity production through greater use of renewables. Some technologies (e.g., PHS, CAES, and VRB and PSB batteries) can eliminate cost-effectively over 90% of CO₂ emissions relative to a no-renewables case. Without energy storage, massive renewable deployment can only achieve about 72% CO₂ emissions reductions (with the base-case 7.0-GW minimum-dispatchability requirement and a \$200 per ton CO₂-emissions tax). In Texas, energy storage deployment yields 57% CO₂-emissions reductions compared to a no-renewables case (assuming an 8.2-GW minimum-dispatchability requirement and a \$200 per ton emissions tax). Without energy storage, 60 GW of renewables reduce emissions by 54% relative to a no-renewables case. Recent analyses [1, 45] show that Texas had over 22 GW of wind installed as of 2017. Thus, the case with 60 GW of renewables represents a significant increase in solar capacity and an already-achieved wind-penetration level.

California has less supply-side flexibility (i.e., more output from nuclear, geothermal, biomass, and hydroelectric units and energy transactions) compared to Texas, resulting in relatively high renewable curtailment in California. Thus, energy storage is valuable in reducing renewable curtailment and displacing fossil-fueled generation. Conversely, even without added renewables, energy storage is cost-effective in Texas with a carbon tax, as it can be used to shift generating loads away from coal-fired units towards natural gas-fired generation.

Our results represent a lower bound on energy storage's role in renewable integration and electricity decarbonization. This is because at high renewable penetrations, energy storage may play other roles that are not captured in our model [3]. For instance, energy storage can be a low-cost source of flexibility to accommodate subhourly or minute-to-minute variability in wind and solar availabilities. Because our model assumes an hourly temporal resolution, such a benefit of energy storage is not captured.

Our results show that its capital cost is the primary factor in determining the scale at which an energy storage technology is deployed. Even with ambitious renewable penetrations and a high emissions tax, a relatively expensive (but high-efficiency) technology, such as Li-ion batteries, has a limited role to play. Our results suggest, however, that modest reductions in Li-ion-battery costs may increase their deployment. We determine this by examining the 'reduced cost' of energy storage capacity, which is obtained from solving our optimization model. In the context of our model, the reduced cost can be interpreted as indicating how much the capital cost of an uneconomic energy storage technology must change before it is cost-effective to build [46]. Our results show that in scenarios in which Li-ion batteries are not built, capital cost reductions of between \$1 per kWh and \$40 per kWh are sufficient to make the technology economically viable. Given the major reductions in battery-manufacturing costs over the past decade, such cost reductions may be possible. This would mean that energy storage technologies that appear uneconomic in our case study may well be viable in the near future. The reduced costs results for other storage technologies are provided in the Supplementary Information.

Given the wide range of costs for Li-ion, NaS, and PSB batteries that are reported in the literature, we conduct a sensitivity analysis, in which the capital costs of Li-ion batteries are reduced to \$259 per kWh and \$59 per kW, the costs of NaS batteries are increased to \$350 per kW and \$350 per kWh, and the costs of PSB batteries are reduced to \$200 per kW and \$90 per kWh. Table 4 summarizes the impacts of these changed capital costs. Specifically, the table reports changes in renewable curtailments and CO₂ emissions relative to the levels that are achieved with the baseline costs that are in Table 3, as a percentage of the baseline curtailment and emissions impacts of Li-ion, NaS, and PSB. The results that are in the table are for 2012, assuming 20 GW of wind and 40 GW of solar are added to each system, a \$200 per ton CO₂-emissions tax, and the base-case minimum-dispatchability requirement for each system. The amounts of energy storage added, renewable curtailments, and CO₂ emissions that are achieved in other scenarios are provided in the Supplementary Information.

Our results demonstrate that increasing the CO₂-emissions tax makes energy storage more cost effective. Yong and McDonald [47] show that an emissions-tax regime that is set by a government with a willingness to commit to it, has a positive influence on the size and the direction of firm-level investment in clean technologies. Thus, adding a strong emissions tax to the already-established energy storage mandate in California may have beneficial economic, policy, and technology-development impacts. **We also show that greater generator flexibility, which is represented through a lower minimum-dispatchability requirement, reduces renewable curtailment and the amount of energy storage that is needed.**

There are some important limitations of our analysis that can be examined in future research. The only environmental impact of electricity production and energy storage use that we examine is CO₂ emissions. **There may, however, be other important impacts. Our results show that PHS holds great promise, due to its relatively low cost.** There are, however, concerns around other environmental impacts of PHS, such as land and water use, species mortality, and impacts on biological production. Moreover, PHS is location-dependent and requires sites with specific topological and geological characteristics [12]. The deployment of CAES is also limited, as specific underground formations are needed to store the compressed air [12]. Further examination of these limitations would provide a more comprehensive understanding of the deployment potential of these technologies.

Our optimization model could be applied to other case studies, with different generation mixes. We assume no degradation of energy storage throughout its operation. **Arbabzadeh et al. [38] show that its degradation does not significantly change the environmental impacts of using energy storage for generation-shifting. Nevertheless, future work could examine the impact of such degradation on the cost-effectiveness of using energy storage for alleviating renewable curtailment.** We also assume that energy storage can operate between 0% and 100% state of charge. **Future analyses can define technology-specific operational windows for energy storage.**

REVIEWERS' COMMENTS:

Reviewer #3 (Remarks to the Author):

The authors have addressed each of the previous sets of reviewer comments well and have greatly enhanced the text. This work provides a number of important contributions to the literature, particularly in demonstrating that energy storage, while potentially offering other value, will clearly reduce renewable curtailment in multiple (and quite diverse) geographic regions and has the potential to impact decarbonization in an environmentally-beneficial way. This work quantifies this and the geographic scope helps to support the conclusions derived by the authors.

In terms of clarity of presentation, the figures are more easily interpretable now, and the results stand out. The document flows well, and the key takeaways are highlighted throughout much more clearly than previous versions. I leave it to the editorial team to determine the proper distribution of figures between the main document and any supplemental information, given journal policy.

The inclusion of the optimization code to support his paper is an important addition for reproducibility, and also significantly enhances the papers ability to achieve greater impact for the research community.

In its current form, I would recommend this paper for acceptance.

Reviewer #3 (Remarks to the Author):

The authors have addressed each of the previous sets of reviewer comments well and have greatly enhanced the text. This work provides a number of important contributions to the literature, particularly in demonstrating that energy storage, while potentially offering other value, will clearly reduce renewable curtailment in multiple (and quite diverse) geographic regions and has the potential to impact decarbonization in an environmentally-beneficial way. This work quantifies this and the geographic scope helps to support the conclusions derived by the authors.

In terms of clarity of presentation, the figures are more easily interpretable now, and the results stand out. The document flows well, and the key takeaways are highlighted throughout much more clearly than previous versions. I leave it to the editorial team to determine the proper distribution of figures between the main document and any supplemental information, given journal policy.

The inclusion of the optimization code to support his paper is an important addition for reproducibility, and also significantly enhances the papers ability to achieve greater impact for the research community.

In its current form, I would recommend this paper for acceptance.

We have included the “Code Availability” statement in the manuscript as follows:

The optimization code that supports the analysis within this paper and other findings of this study are available from the corresponding author upon reasonable request.